# Meaningless Tokens, Meaningful Gains: How Activation Shifts Enhance LLM Reasoning

## Abstract

Motivated by the puzzling observation that inserting long sequences of meaningless tokens before the query prompt can consistently enhance LLM reasoning performance, this work analyzes the underlying mechanism driving this phenomenon and based on these insights proposes a more principled method that allows for similar performance gains. First, we find that the improvements arise from a redistribution of activations in the LLM's MLP layers, where near zero activations become less frequent while large magnitude activations increase. This redistribution enhances the model's representational capacity by suppressing weak signals and promoting stronger, more informative ones. Building on this insight, we propose the Activation Redistribution Module (ARM), a lightweight inference-time technique that modifies activations directly without altering the input sequence. ARM adaptively identifies near-zero activations after the non-linear function and shifts them outward, implicitly reproducing the beneficial effects of meaningless tokens in a controlled manner. Extensive experiments across diverse benchmarks and model architectures clearly show that ARM consistently improves LLM performance on reasoning tasks while requiring only a few lines of simple code to implement. Our findings deliver both a clear mechanistic explanation for the unexpected benefits of meaningless tokens and a simple yet effective technique that harnesses activation redistribution to further improve LLM performance. The code has been released at ARM-Meaningless-tokens.

## 1 Introduction

Large language models (LLMs) are known to be sensitive to subtle variations in their inputs, which makes it important to understand how tokens influence predictions (Guan et al., 2025; Errica et al., 2024; Zhuo et al., 2024). In this paper, we present a surprisingly counterintuitive finding named **meaningless-token effect**: inserting **long sequences of meaningless tokens**, such as repeated punctuation or separators, into prompts can consistently improve the performance of LLMs, particularly on reasoning tasks. Contrary to common intuition that long and irrelevant tokens are like noise and thus useless or even harmful during inference (Jiang et al., 2024; Guan et al., 2025), our experiments reveal the opposite. When long sequences of meaningless tokens are appended before query prompts, models that previously struggled with certain problems can produce correct solutions, as illustrated in the left panel of Figure 1 (see more examples in Appendix J). This effect occurs consistently across tasks and models, suggesting a counterintuitive behavior of LLMs pending deeper investigation.

This unexpected result raises fundamental questions about how LLMs process input and what aspects of their internal computation are being affected. *Why should tokens that convey no meaning lead to measurable performance gains? Are they simply acting as noise, or do they restructure representations in a systematic way that supports better reasoning?* To answer these questions, we move beyond surface level observations and conduct a detailed investigation of the mechanisms behind this effect. Our analysis shows that the influence of meaningless tokens arises primarily in the first layer, and their effect on meaningful tokens can be approximated as an affine transformation of the attention outputs. As demonstrated in the middle schematic diagram of Figure 1, the resulting transformation shifts the distribution of activations in the MLP: the proportion of near-zero activations decreases, while more activations are pushed outward toward larger positive and negative values. The rightmost plot in Figure 1 gives a visualization of this process. We hypothesize that redistribution fosters richer exploration, enhancing reasoning performance, and clarify the mechanism by decomposing the

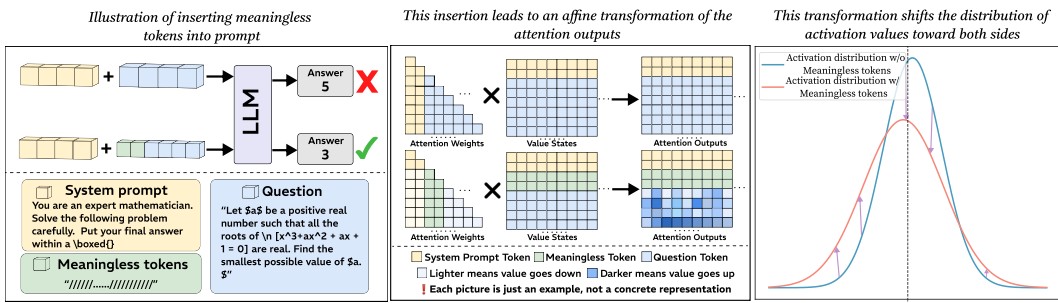

Figure 1: The left panel illustrates how meaningless-token effect can improve model performance. The middle panel shows the changes occurring in the attention module after introducing meaningless tokens. The right panel depicts the redistribution of activations that results from adding these tokens.

transformation into coefficient and bias terms. Our theoretical analysis shows how each component shapes activation variance and induces the observed distributional shift.

Building on these insights, we propose **ARM** (an **A**ctivation **R**edistribution **M**odule), a lightweight alternative to explicit meaningless-token insertion. ARM requires only a few lines of code modification and no additional training. It automatically identifies a proportion of near-zero activations after the non-linear function and shifts their values outward, yielding a smoother and less sparse activation distribution. In doing so, ARM reproduces the beneficial effects of meaningless tokens without altering the input sequence and consistently improves LLM performance on reasoning and related tasks. In summary, the key findings and contributions of our work are:

- We uncover a **meaningless-token effect in LLMs**: inserting meaningless tokens, far from being harmful, systematically improves reasoning in LLMs. This runs counter to the common assumption that such tokens only add noise.

- Through theoretical and empirical analysis, we show that these tokens induce an **activation redistribution** effect in the first-layer MLP, reducing near-zero activations and increasing variance.

- Building on this understanding, we present ARM, a lightweight inference-time instantiation to demonstrate that the phenomenon can be directly harnessed.

## 2 OBSERVATION: INSERTING MEANINGLESS TOKENS INDUCES AN AFFINE TRANSFORMATION ON MEANINGFUL TOKEN REPRESENTATIONS

We observe that meaningless tokens, such as a sequence of slashes ("/") with appropriate lengths can enhance the performance of LLMs, particularly on reasoning tasks[1]. As shown in Table 1, when we insert a fixed-length sequence of meaningless tokens between the system prompt and the question, all evaluated models exhibit performance improvements on Math-500 and AIME 2022-2024 to different degrees. This consistent improvement suggests that the inserted meaningless tokens are not simply ignored or detrimental to the models; rather, they exert a positive influence, likely through non-trivial interactions with the models' internal representations. To investigate this phenomenon, we start our analysis from the attention module. The formula of attention is:

Table 1: Performance on mathematical reasoning datasets with and without meaningless tokens across different models. "w/o" denotes the absence of meaningless tokens, while "w/" denotes their presence. We test each model five times to get the average result. red means corresponding meaningless words and its number.

| Methods | MATH-500 | | AIME22-24 | |
|---|---|---|---|---|
| | w/o | w/ | w/o | w/ |
| Qwen2.5-Math-1.5B | 63.9 | **65.9**$_{55/}$ | 14.4 | **17.5**$_{70/}$ |
| Qwen2.5-Math-7B | 72.3 | **74.6**$_{55/}$ | 23.1 | **23.3**$_{55/}$ |
| DS-R1-Qwen-7B | 52.7 | **53.1**$_{10/}$ | 3.2 | **4.4**$_{70/}$ |
| DS-Math-7B-instruct | 39.5 | **42.1**$_{55?}$ | 7.8 | **12.3**$_{10?}$ |
| Llama-3.1-8B-Instruct | 41.8 | **42.1**$_{55/}$ | 7.9 | **9.9**$_{70/}$ |
| Qwen-2.5-32B-Instruct | 81.3 | **81.7**$_{10?}$ | 17.6 | **22.8**$_{70/}$ |

$\text{Attention}(Q, K, V) = \text{softmax}\left(\frac{QK^\top}{\sqrt{d_k}}\right) V$, where $Q, K, V$ are query vectors, key vectors and value vectors respectively, $d$ is the dimensionality of key/query. From this equation, adding extra tokens

---

[1]Varying token length, type, and position affects performance, as shown in Appendix F.

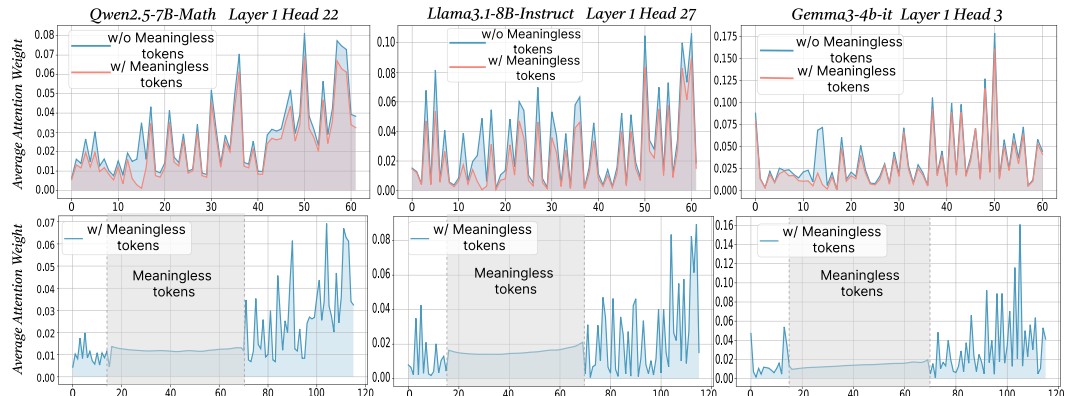

Figure 2: The x-axis shows token indices. The top row of Figure 2 shows how the model's attention to meaningful tokens changes after inserting meaningless tokens, where red denotes the attention after insertion and blue denotes the attention before insertion. For visualization clarity, we omit the meaningless tokens in this view. The bottom row of Figure 2 presents the attention received by the meaningless tokens themselves, which is consistently low across positions. We show additional average attention weights in Appendix I and layer-wise analyses in Section F.4.

introduces additional terms into the softmax normalization, enlarging the softmax normalization denominator. Although the new tokens typically receive small weights, their presence redistributes probability mass and reduces the relative share of attention allocated to the original tokens. To probe the underlying case, we directly compare input's attention weights with and without meaningless tokens while keeping tokens indices aligned in the first layers. For every token we computed the mean of its column below the diagonal of the attention matrix to measure the extent to which each token receives attention from all downstream tokens (Bogdan et al., 2025). When a string of meaningless tokens are present, the model assigns only small weights to each token, intuitively indicating that the model only pays little attention to them (see Figure 2 bottom row). The top row of Figure 2 presents a direct comparison of the attention to meaningful tokens without (blue) or with meaningless tokens (red; meaningless token indices are removed from visualization to allow for direct comparison). Among meaningful tokens, the average attention is decreased in the meaningless-token condition, especially driven by decreased high-attention spikes. The attention weights of the original prompt after inserting meaningless tokens are: $W' = \lambda \cdot \text{softmax}\left(\frac{QK^\top}{\sqrt{d_k}}\right)$, where $W_{attn}$ are the attention weights after softmax, and $\lambda$ is the drop percentage of attention weights in the original prompt after adding meaningless tokens. Then, the attention output for each token not only obtains the weighted combination of the original tokens, but also includes attention weights and values from the meaningless tokens. Thus, the attention output can be expressed as:

$$\text{Attn\_Output}_{new} = W'_j V_j + W_i V_i, \tag{1}$$

where Attn_Output corresponds to the output of attention mechanism for each token in the original prompt, $W'_j$ and $V_j$ are the attention weight and value vectors of the original prompt, and $W_i$ and $V_i$ are the attention weight and value vectors of meaningless tokens. As the meaningless tokens are repeated in long sequences and contribute no semantic information, the values of these tokens are identical, and their attention weights are small in a similar magnitude. Therefore, as shown in Equation 1, the term $W_i V_i$ primarily shifts the final attention output along an approximately unified direction as they accumulate, without introducing diverse semantic components. In this formula, $W_j V_j$ is the value of original attention output, we see $W_i V_i$ as $\Sigma_\sigma$. As a result, the attention output of meaningful tokens after adding meaningless tokens can be

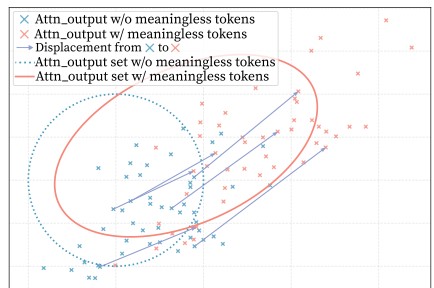

Figure 3: After adding meaningless tokens, each token vector is affinely transformed: blue points show the original vectors, and red points show them after the addition. Arrow is change direction. x axis and y axis are the first two main component of PCA. This figure show the relative relationship before and after affin transformation.

seen as an affine transformation expressed as:

$$\text{Attn\_Output}_{new} = \lambda \cdot \text{Attn\_Output} + \Sigma_\sigma, \tag{2}$$

where Attn_Output is $W_j V_j$. Following this equation, the introduction of meaningless tokens transforms the attention output of meaningful tokens into an affine function, consisting of a scaled original term ($\lambda \cdot$ Attn_Output) and an additional bias ($\Sigma_\sigma$). Figure 3 illustrates the process of this transformation. After the attention module the affine transformed output passes through RMSNorm and serves as the input to the MLP. In the next section, we examine in detail how this transformation propagates through the subsequent MLP layers and shapes the model's overall activation distribution.

## 3 ANALYSIS: WHY AFFINE TRANSFORMATION IMPROVE REASONING PERFORMANCE

Having established in the previous sections that meaningless-token effect induces scaling and bias terms that produce an affine transformation of the attention output, we next examine how this transformation propagates through the subsequent MLP modules and affects reasoning. In Equation 2, we decompose the transformation's influencing factors into two primary components: the **scaling factors** $\lambda$ controls the magnitude of activations, and the **bias factors** $\Sigma_\sigma$, a bounded zero-mean bias term reflecting the variation in attention outputs before and after meaningless-token insertion which introduce structured shifts in the activation distribution. Together, these two factors determine how the transformed attention representations shape the dynamics of the MLP layers.

### 3.1 AFFINE TRANSFORMATION INFLUENCE THE OUTPUT OF GATE LAYER

> **Key Takeaway**
>
> We demonstrate that applying an affine transformation, through both scaling and bias factors, systematically increases the variance of the gate layer's output.

In this part, we show that these two factors increase the gate projection layer variance in MLP layer. As discussed above, because these tokens have low attention weights and nearly identical values, they shift the RMSNorm input almost entirely along a single direction with small margin; consequently, RMSNorm largely absorbs this change, producing only a minor numerical adjustment without adding semantic information. Specifically, the two factors act through different mechanisms. For the **scaling factors**, before entering the MLP, the attention output undergoes output projection and residual connection, which can be written as $x(\lambda) = \text{res} + \lambda * \text{U} * \text{A}$, where A is the attention output and U the projection weights. Treating $\lambda$ as a functional variable, the RMSNorm output becomes $y(\lambda) = \text{RMS}(x(\lambda))$. For the $j$-th gate dimension, $z_j(\lambda) = w_j^\top y(\lambda)$, and a small variation $\Delta\lambda$ leads to the variance change of this dimension.

$$\text{Var}[z_j(\lambda + \Delta\lambda)] = \text{Var}[z_j(\lambda)] + 2\text{Cov}(z_j(\lambda), g_j(\lambda))\Delta\lambda + \text{Var}[g_j(\lambda)]\Delta\lambda^2, \tag{3}$$

the third term in Equation 3 remains strictly positive for all admissible parameters. Moreover, as $\Delta\lambda$ increases, this term exhibits monotonic growth and asymptotically dominates the second term, thereby guaranteeing a strictly increasing overall variance. We analyze the range of $\Delta\lambda$ in Appendix E. In the case of **bias factors**, we model the perturbation as stochastic noise which is bounded, zero-mean and statistically independent from the original attention output across all dimensions, which contributes an additional variance component and interacts non-trivially with the subsequent RMSNorm operation. Formally, after noise injection, the RMSNorm input can be written as $x = x_0 + W\Sigma_\sigma$, where $W$ is the linear coefficient of matrix $x$ preceding RMSNorm. After normalization, the covariance of the output can be expressed as:

$$\text{Cov}(y) = J_q\text{Cov}(x)J_q^\top + o(\|x - x_0\|^2) \tag{4}$$

where $x_0$ is the mean expansion point, $J_q$ is the Jacobian matrix of the RMSNorm mapping. Since the variance of the added perturbation is very small, the higher-order terms can be disregarded. In this case, the bias factor will bias the input of RMSNorm and lead to an increase in the covariance $\text{Cov}(y)$. Subsequently, the input to the activation function can be written as $z = W_{gate}(x + W\Sigma_\sigma)$. Based on the properties of the covariance, the variance of the $j$-th dimension is given by:

$$\text{Var}[z_j] \approx e_j^\top W_{gate}[J_q\text{Cov}(x)J_q^\top]W_{gate}^\top e_j, \tag{5}$$

since the projection of the vector onto the tangent space is almost never zero in LLMs' high dimensions, the resulting variance must be strictly greater than zero. From this, we can deduce that these two factors increase the variance of the output. In general, the scaling factors increase variance by amplifying inter-sample differences, whereas the bias factors correspondingly increase variance by enlarging the covariance structure across dimensions.

## 3.2 VARIANCE CHANGE LEADS TO ACTIVATION REDISTRIBUTION

> **Key Takeaway**
>
> Our analysis shows that an increase in the input variance of activation functions broadens and reshapes the output activation distribution by raising both its mean and its variance.

As the variance of gate layer outputs grows under perturbations, the subsequent activation function further reshapes these signals by compressing values near zero. This motivates redistributing near-zero activations. For each sample in the hidden state, the second-order Taylor expansion on $\phi$, the activation function output is:

$$\phi(\mu + \sigma) = \phi(\mu) + \phi^{'}(\mu)\sigma + \frac{1}{2}\phi^{''}(\mu)\sigma^2 + o(|\sigma|^3), \tag{6}$$

where $\sigma$ can represent both $\Delta k$ in scaling factor and $\Sigma_\sigma$ in bias factor. We denote the input to the activation function as $z = \mu + \sigma$. For the $j$-th dimension of the hidden state, the expectation and variance of the activation output can be expressed as:

$$\mathbb{E}[\phi(z_j)] = \mathbb{E}[\phi(\mu_j)] + \mathbb{E}[\phi^{'}(\mu_j)\sigma] + \mathbb{E}[\frac{1}{2}\phi^{''}(\mu_j)\sigma^2] + o(\mathbb{E}|\sigma|^3), \tag{7}$$

$$\mathrm{Var}[\phi(z_j)] = \phi^{'}(\mu_j)^2\mathrm{Var}_j + o(\mathrm{Var}_j^2). \tag{8}$$

From above equations, We infer that distributional changes map to variations in expectation and variance. On a single dimension, activations shift in both directions; from Equation 6, higher-order terms are negligible, and the first derivative of GeLU/SiLU near zero is positive. Since perturbations include both signs, extrapolated activations also fluctuate around zero. From Equation 7, $\mathbb{E}[\sigma^2] = \mathrm{Var}_j$. For the bias factor, the zero-mean perturbation removes the first-order term. For scaling factors, expanding at the population mean gives $\mathbb{E}[\phi^{'}(z_j)g_j] = 0$, again canceling the first order. The second derivative near zero is strictly positive. From Equation 8, $\mathrm{Var}_j$ increases, and so does the activation histogram variance, as the function is nearly linear near zero. In summary, scaling and bias factors jointly enlarge activation variance, expressed as:

$$\mathrm{Var}_j \approx \mathbb{E}[\mathrm{Var}_j^{(\Sigma_\sigma)}] + \mathrm{Var}(g_j^\lambda). \tag{9}$$

The first term represents the expected variance of the $j$-th hidden states under the influence of the bias factor. Since the bias factor varies across individual cases, taking the expectation is necessary to capture its overall impact. The second term corresponds to the variance induced by scaling factors, which inherently reflects the aggregate change. When combining them, the overall variance of the outputs of nonlinear activation functions increases, the mean shifts upward, and the activation distribution becomes broader, manifested as heavier tails and a thinner center. More details of

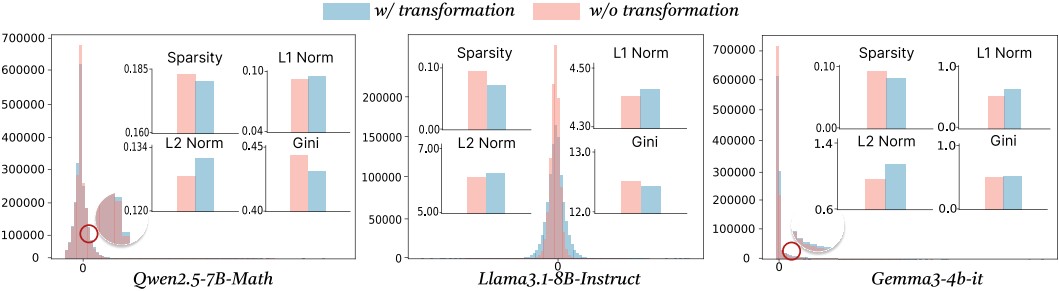

Figure 4: The histogram of the frequency of activations after activation functions in MLP, the sub-figure is the comparison of 4 metrics between before and after transformation.

above analysis and relative proof are in Appendix E. Moreover, we presume the reason that this redistribution has a positive impact on reasoning tasks is that reasoning-critical tokens (digits, operators, conjunctions) have a higher fraction of near-zero activations. Elevating their activation levels strengthens their representations and improves reasoning performance; see Section 6 for details.

### 3.3 VERIFICATION OF ACTIVATION REDISTRIBUTION

To verify whether the activation redistribution pattern in Section 3.2 indeed occurs in LLMs, Figure 4 illustrates the activation distribution after the first-layer MLP, explicitly comparing states before and after the transformation defined in Equation 2. We also comprehensively assess the transformation of activation states using several quantitative indicators, including:

- **Relative Sparsity**: Defined as the proportion of activations after the transformation whose values fall below the pre-transformation threshold.

- **L1 Norm**: The sum of the absolute activation values; smaller values indicate higher sparsity.

- **L2 Norm**: A measure of the overall magnitude of activations.

- **Gini Coefficient**: An indicator of the smoothness of the histogram distribution, where smaller absolute values correspond to smoother distributions.

From Figure 4, we observe that after transformation, the frequency of near-zero activations decreases, while the frequency of absolute high-magnitude activations increases. Both sparsity and smoothness in the activation distribution are improved. Specifically, the relative sparsity consistently decreases across all three models while the L1 and L2 norms increase, aligning with the previous phenomenon.

## 4 METHOD: ACTIVATION REDISTRIBUTION MODULE

Inspired by the previous finding that meaningless tokens can shift meaningful activations and boost LLM performance, we propose ARM—a simple method replacing explicit meaningless tokens with an implicit mechanism that adjusts the MLP activation distribution after the activation function. Our approach has two steps: First, adaptively identify a proportion of near-zero activations based on the model and input; Then, extrapolate them outward to redistribute the activation pattern. The top half of Figure 5 shows the first-layer MLP with ARM, where selected activations around zero are shifted outward, reducing their frequency and increasing larger-magnitude activations. The bottom half of Figure 5 presents the ARM-specific code, a lightweight function inserted into the first-layer MLP without affecting inference speed.

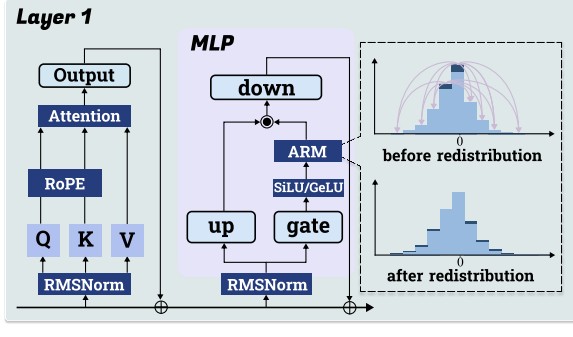

```
def forward(x, layer_idx): # in first layer
    activation = self.act_fn(self.gate_proj(x))
    #Our function
    activation_alter = self.arm(activation.clone())
    down_proj = self.down_proj(activation_alter *
        self.up_proj(x))
    return down_proj
```

Figure 5: The upper panel illustrates the first-layer LLM architecture with ARM, while the lower panel presents the corresponding ARM code in the MLP module.

As shown in Appendix D, ARM's time complexity is negligible within the MLP context. **The significance of the ARM method is twofold**. Firstly, it adds further evidence deductively supporting our theoretical analysis in Section 3. By directly replacing explicit meaningless token insertion with implicit activation redistribution, ARM yields a similar improvement in reasoning across models and benchmarks, thus strengthening our theoretical framework. Secondly, we introduce ARM as a lightweight inference time trick for boosting reasoning, which is not only robustly effective on its own (see experiments in Section 5) but also compatible with existing inference time scaling methods (see Appendix G.3).

Table 2: After adding ARM to the first-layer MLP, we report reasoning-task performance for six models, using a dash ('–') for accuracies below 5% to indicate incapability.

| Model | Setting | GPQA Diamond | Math-500 | AIME 2022-2024 | AIME 2025 | LiveCodeBench | Humaneval |
| --- | --- | --- | --- | --- | --- | --- | --- |
| | | Pass@1 | Pass@1 | Pass@1 | Pass@1 | Pass@1 | Pass@1 |
| Qwen2.5 Math-1.5B | Baseline | 27.3 | 63.8 | 14.4 | 6.7 | - | 6.1 |
| | ARM | 28.8 | 67.0 | 18.9 | 10.0 | - | 8.5 |
| | Improve Rate (%) | 1.5↑ | 3.2↑ | 4.5↑ | 3.3↑ | - | 2.4↑ |
| Qwen2.5 Math-7B | Baseline | 30.3 | 72.4 | 23.3 | 10.0 | - | 15.2 |
| | ARM | 34.9 | 73.4 | 25.6 | 13.3 | - | 17.7 |
| | Improve Rate (%) | 4.6↑ | 1.0↑ | 2.3↑ | 3.3↑ | - | 2.5↑ |
| Qwen2.5 7B-Instruct | Baseline | 28.3 | 73.2 | 20.0 | 10.0 | 29.7 | 43.9 |
| | ARM | 29.8 | 75.2 | 20.0 | 23.3 | 31.9 | 47.6 |
| | Improve Rate (%) | 1.5↑ | 2.0↑ | 0 | 13.3↑ | 2.2↑ | 3.7↑ |
| Qwen2.5 32B-Instruct | Baseline | 35.4 | 82.6 | 16.7 | 20.0 | 49.5 | 50.0 |
| | ARM | 35.9 | 82.6 | 18.8 | 26.7 | 49.5 | 51.2 |
| | Improve Rate (%) | 0.5↑ | 0 | 2.1↑ | 6.7↑ | 0 | 1.2↑ |
| Llama3.1 8B-Instruct | Baseline | 28.3 | 43.0 | 11.1 | - | 11.9 | 45.7 |
| | ARM | 31.3 | 45.8 | 13.3 | - | 17.0 | 47.6 |
| | Improve Rate (%) | 3.0↑ | 2.8↑ | 2.2↑ | - | 5.1↑ | 1.9↑ |
| Gemma3 4b-it | Baseline | 34.3 | 72.6 | 13.3 | 20.0 | 20.2 | 17.1 |
| | ARM | 35.9 | 74.0 | 17.8 | 23.3 | 20.6 | 20.7 |
| | Improve Rate (%) | 1.5↑ | 1.4↑ | 4.5↑ | 3.3↑ | 0.4↑ | 3.6↑ |
| Gemma3 27b-it | Baseline | 33.3 | 85.4 | 25.6 | 26.7 | 31.9 | 9.1 |
| | ARM | 33.8 | 86.2 | 31.1 | 30.0 | 34.2 | 11.6 |
| | Improve Rate (%) | 0.5↑ | 0.8↑ | 4.4↑ | 3.3↑ | 2.3↑ | 2.5↑ |

## 4.1 SELECT APPROPRIATE CHANGE PROPORTION

Our method first selects a proportion of activations to be modified. However, different models exhibit varying sensitivities to meaningless tokens. To address this, we propose a dynamic strategy that adjusts the fraction of near-zero activations to be altered during inference. To determine this proportion, we measure the dispersion of activations around zero. Specifically, we define a neighborhood $\epsilon$ based on the activation distribution to identify which activations are considered "close to zero". We adopt the Median Absolute Deviation (MAD) as our dispersion metric, since MAD is robust to outliers and better captures the core distribution. The threshold $\epsilon$ is given by: $\epsilon = \kappa * \mathrm{MAD} * c$, where $\kappa$ is a consistency constant, $c$ is a hyperparameter controlling the width of the near-zero range. Next, we compute the fraction of activations falling within $[-\epsilon, \epsilon]$ This fraction $p$ represents the proportion of activations that we think to be near zero. As a result, the fraction we want to change is fraction $= clip(p, (p_{\min}, p_{\max}))$. Here, $p$ denotes the calculated fraction, while $p_{\min}$ and $p_{\max}$ serve as bounds to prevent the scale from becoming either too small or excessively large. In our experiments, we set $p_{\min} = 0.02$ and $p_{\max} = 0.25$.

## 4.2 REDISTRIBUTION OF ACTIVATION VALUES

After selecting the elements, we preserve its sign and adjust only its magnitude. Specifically, we add a positive or negative value depending on the element's sign. To constrain the modified values within a reasonable range, the range is defined as follows:

$$\mathrm{R} = \begin{cases} [0, \mathrm{Q}_{p_1}(\mathrm{Activations})], & \mathrm{sign} = 1, \\ [\min(\mathrm{Activations}), 0], & \mathrm{sign} = -1. \end{cases} \tag{10}$$

Where R is the range of modified values. In this range, we set the lower bound to the minimum activation value when sign $= -1$, since activation functions such as SiLU and GeLU typically attain their smallest values on the negative side. For the upper bound when sign $= 1$, we select the value corresponding to the $p_1$-th percentile of the activation distribution. Here, $p_1$ is a hyperparameter that depends on the distribution of activations. $\mathrm{Q}_{p_1}(\mathrm{Activations})$ is the upper bound when we changing the chosen activations. The value of $p_1$ depends on the distribution of activations and the value of $c$. Finally, we generate a random value in R and add it to the activation in order to modify its value. In this way, we adaptively adjust an appropriate proportion of activations, enriching the distribution with more effective values. We shows how to choose hyperparameter in Appendix H.

## 5 EXPERIMENTS

We evaluate our method on reasoning and non-reasoning tasks using seven models: Qwen2.5-Math-1.5B, Qwen2.5-Math-7B, Qwen2.5-Instruct-7B, Qwen2.5-Instruct-32B (qwe, 2025), Llama3.1-8B-Instruct (gra, 2024), Gemma3-4b-it, and Gemma3-27b-it (gem, 2025). All models use default generation parameters. For reasoning tasks, we cover three skill areas: (1) General: GPQA (Rein et al., 2024), a challenging expert-authored multiple-choice dataset; (2) Math & Text Reasoning: MATH-500 (Lightman et al., 2023), AIME'24 (AIME, 2024), and AIME'25 (AIME, 2025); (3) Agent & Coding: LiveCodeBench (Jain et al., 2024) and HumanEval (Chen et al., 2021). For non-reasoning tasks, we use GSM8K (Cobbe et al., 2021), ARC-E (Clark et al., 2018), ARC-C (Clark et al., 2018), MMLU (Hendrycks et al., 2021), BoolQ (Clark et al., 2019), HellaSwag (Zellers et al., 2019), and OpenBookQA (Mihaylov et al., 2018).

### 5.1 EXPERIMENT RESULTS ANALYSIS

For **reasoning tasks**, the results in Table 2 show pass@1 accuracy across multiple benchmarks. Our method consistently improves performance across most models and datasets, with the effect more pronounced in smaller models (e.g., Qwen2.5-Math-7B shows larger gains than Qwen2.5-32B-Instruct). On challenging benchmarks, however, improvements are limited when models lack sufficient capacity or when baseline accuracy is near saturation. For **non-reasoning tasks** (see Table 3b), applying ARM to the first-layer MLP yields little change. We attribute this to their largely factual nature, where models already have the necessary knowledge and response formats, requiring minimal reasoning. By contrast, for reasoning tasks, altering early activations helps reorganize knowledge, strengthens intermediate representations, and facilitates more effective and consistent reasoning.

### 5.2 COMPARISON OF MEANINGLESS TOKENS AND ARM

In Table 3a, we provide a direct comparison between our proposed ARM method and the strategy of inserting a suitable number of meaningless tokens. The results demonstrate that both approaches are capable of improving model performance and neither requires post-training, therefore presenting lightweight interventions that lead to robust performance gains. However, since ARM directly utilizes the fundamental principle driving the meaningless-token effect, it provides more stable results. While the meaningless-token effect is pervasive, our experiments show that the effect itself depends heavily on the specific choice of token length and placement, and thus may be unstable or difficult to generalize across tasks. ARM provides a more principled and model-internal mechanism that directly reshapes the activation distribution within the MLP, yielding more consistent gains without relying on heuristic token engineering. In sum, while the insertion of a meaningless token string on the prompt level might seem like a promising prompt-tuning adjustment on the surface, it comes with an instability of the effect which ARM eliminates. This contrast highlights the trade-off between ease of use and robustness, and further underscores the value of ARM as a systematic method for enhancing the reasoning ability in large language models.

### 5.3 EXPLORATION CAPABILITIES AFTER ARM

As discussed earlier, we hypothesize that redistributing activations enables the model to explore the reasoning space more effectively. To test this hypothesis, we evaluate the model's pass@3

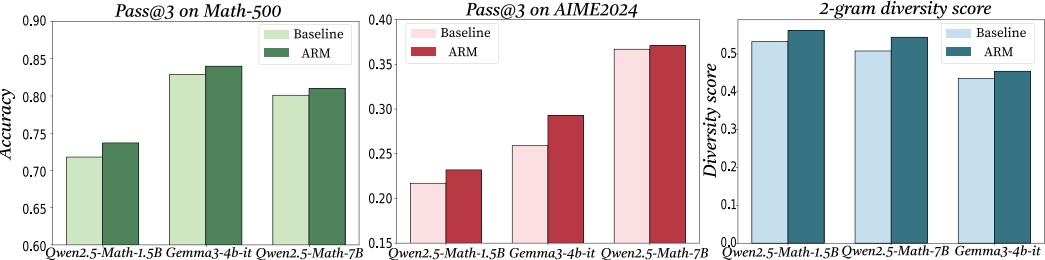

Figure 6: The first two figures show pass@3 on Math-500 and AIME 2022-2024 for three models with and without ARM, and the last shows their 2-gram diversity under both conditions.

Table 3: Table (a) compares the performance of meaningless tokens and ARM, and Table (b) reports ARM's results on non-reasoning tasks.

(a) Pass@1 on Math-500 and AIME 2022-2024 with meaningless tokens (Mless) or ARM.

| Model | Setting | Math-500 | AIME 2022-2024 |
|---|---|---|---|
| Qwen2.5 Math-7B | Baseline | 72.4 | 23.3 |
| | Mless | **75.0** | 24.4 |
| | ARM | 73.4 | **25.6** |
| Llama3.1 8B-Instruct | Baseline | 43.0 | 11.1 |
| | Mless | 44.9 | **13.3** |
| | ARM | **45.8** | **13.3** |

(b) Performance of models with ARM on non-reasoning tasks. Additional results are in Appendix G.

| Model | Setting | GSM8K | ARC-E | HellaSwag |
|---|---|---|---|---|
| Qwen2.5 Math-1.5B | Baseline | 78.0 | 39.3 | 39.1 |
| | ARM | 78.6 | 39.3 | 39.5 |
| | **Improve Rate (%)** | **0.6↑** | 0 | **0.4↑** |
| Llama3.1 8B-Instruct | Baseline | 80.0 | 46.6 | 56.8 |
| | ARM | 82.4 | 47.1 | 57.3 |
| | **Improve Rate (%)** | **2.4↑** | **0.5↑** | **0.5↑** |

performance on the Math-500 and AIME 2022-2024 benchmarks as well as its 2-gram diversity. As shown in Figure 6, applying activation redistribution consistently yields higher pass@3 scores compared to the baselines on both tasks. In addition, the 2-gram diversity under ARM is also greater than that without ARM. These findings indicate that activation redistribution not only improves the likelihood of arriving at correct solutions within multiple samples but also promotes more diverse reasoning paths. This dual effect suggests that ARM enhances both the effectiveness and the breadth of the model's internal reasoning processes, reinforcing our hypothesis that carefully manipulating internal activations can expand a model's reasoning capacity without additional training or parameter growth.

## 6    DISCUSSION: WHY ACTIVATION REDISTRIBUTION ENHANCES LLM REASONING PERFORMANCE

We provide one possible explanation for why redistributing the near-zero activations can improve the reasoning performance of LLMs. We categorize all tokens in Math-500 into four classes: digits, conjunctions, operators, and other tokens. For each class, we compute the average proportion of activations falling within near-zero range, which reflects how many dimensions of the hidden representation remain nearly inactive. The results are presented in Figure 7. As shown, normal tokens exhibit the lowest near-zero proportion, while digits, operators, and conjunctions show substantially higher proportions, which means that in the high-frequency near-zero activations after activation function, a larger portion of them are derived from

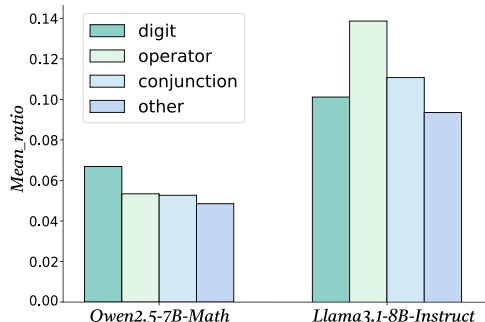

Figure 7: Percentage of near-zero activations across the four token types in the Math-500 dataset.

these tokens. This suggests that although these tokens are crucial for reasoning, their information is insufficiently activated by the model. Our observation is consistent with the findings of Huan et al. (2025), which highlight the increasing importance of conjunctions after reinforcement learning, and also aligns with the recognized role of digits and operators in reasoning tasks such as mathematics and coding. Consequently, redistributing activations around zero enhances the representation of under-activated yet semantically important tokens, improving reasoning performance.

## 7    RELATED WORK

Recent studies notice that symbols in an LLM's input may affect their internal mechanism. Sun et al. (2024) show large activations for separators, periods, or newlines, suggesting that these tokens carry model biases. Razzhigaev et al. (2025) find that commas are essential for contextual memory, while Chauhan et al. (2025) and Min et al. (2024) highlight punctuation as attention sinks, memory aids, and semantic cues. Moreover, Chadimová et al. (2024) show that replacing words with meaningless tokens can reduce cognitive biases, whereas Li et al. (2024) report that such "glitch tokens" may also cause misunderstandings, refusals, or irrelevant outputs. Our work adds explanation to the puzzling downstream benefits that the inclusion of a string of meaningless tokens contributes to reasoning performance and shows how deep investigations of the underlying mechanisms can lead to improved inference solutions. **We provide an extended discussion of related works in Appendix B.**

## 8 CONCLUSION

In this paper, we report a meaningless-token effect that inserting long sequences of meaningless tokens improves model performance, particularly on reasoning tasks. Our analysis suggests that it stems from the fact that meaningless tokens induce an affine transformation on meaningful tokens, thereby redistributing their activations and enabling key information to be more effectively utilized. Building on this insight, we introduce ARM, a lightweight and training-free method for activation redistribution, which strengthens our analysis and serves as a practical approach for consistently improving LLM performance on reasoning tasks.

## ETHICS STATEMENT

All datasets used in this work are publicly available and contain no sensitive information. Our method enhances LLM reasoning without introducing new data collection or human interaction. While stronger reasoning ability may be misused, we emphasize that this work is intended for beneficial research and responsible applications.

## REPRODUCIBILITY STATEMENT

We will release our code and data once the paper is published. The appendix includes detailed experimental setups and hyperparameters so that others can reproduce our results. We also encourage the community to follow good research practices when using our code and data, to help maintain the reliability and transparency of future work.

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

CONTENTS

## A  DISCLOSURE OF LLM USAGE

This paper used LLMs to assist with grammar checking.

## B  RELATED WORK

### B.1  MEANINGLESS TOKENS IN LLMS

Recent studies have shown that seemingly meaningless tokens, such as punctuation marks, play a non-trivial role in information propagation and reasoning within large language models (LLMs). For example, Sun et al. (2024) report that LLMs exhibit large activations in response to separators, periods, or newline characters, suggesting that these tokens can serve as carriers of model biases. Similarly, Razzhigaev et al. (2025) demonstrate that tokens such as commas act as crucial elements in maintaining contextual memory: removing them significantly degrades performance on context-understanding tasks. Chauhan et al. (2025) further argue that punctuation may function as attention sinks or assist the memory mechanism, while Min et al. (2024) highlight its value in semantic construction, enabling models to better capture contextual structure. In addition, Chadimová et al. (2024) show that substituting certain words with meaningless tokens can mitigate cognitive biases in LLMs. Conversely, Li et al. (2024) illustrate that meaningless "glitch tokens" can induce misunderstandings, refusals, or irrelevant generations. However, these works primarily examine the effects of individual meaningless tokens, without considering the broader impact of longer meaningless token sequences.

More recently, several studies have explored the role of long meaningless token sequences and reported their surprising positive influence on LLM performance. For instance, Zhou et al. (2024) find that appending meaningless tokens to the end of prompts can trigger or defend against jailbreak behaviors. Duan et al. (2025) also search meaningless tokens to improve model's performance. Similarly, Shi et al. (2024) show that adding long meaningless sequences after a sentence can improve model performance on certain tasks. Pfau et al. (2024), Duan et al. (2025) and London & Kanade (2025) report that substituting meaningful tokens with filler-like tokens (e.g., '. . .') in the training data preserves the model's ability to solve questions, suggesting that even without meaningful tokens the model can perform implicit computation. Meanwhile, there are also some methods to improve the reasoning performance of LLMs (Dhanraj & Eliasmith, 2025; Højer et al., 2025; Sheng et al., 2025). Herel & Mikolov (2024); London & Kanade (2025); Kim et al. (2025); Goyal et al. (2023) analyze the effectiveness of thinking tokens which are meaningless before fine-tuning. Despite these empirical findings and methods, there is still a lack of systematic analysis explaining why meaningless tokens, especially in longer sequences, can play such a counterintuitive yet beneficial role in shaping LLM reasoning behavior.

### B.2  ACTIVATIONS ANALYSIS IN LLMS

Activation analysis is a popular method for explaining the mechanics of LLMs (Wang et al., 2025; Kawasaki et al., 2024; Pham & Nguyen, 2024; Rai & Yao, 2024). Owen et al. (2025) supplement Sun et al. (2024) by analyzing the activations after MLP to study how massive values influence bias and large attention. Wang et al. (2025) test hidden states across all layers to examine the importance of parameters in different layers. Zhao et al. (2025) use activations to determine whether an attention head is activated after training. Kaul et al. (2024) analyze attention activations and find that almost all activations focus on the first tokens; they also analyze high activations in the FFN. Luo et al. (2024) systematically study the magnitude law and influencing factors of activation sparsity in decoder Transformer architectures, showing that different activation functions (ReLU vs. SiLU) lead to drastically different sparsity trends during training. In Liu et al. (2024), activation refers to the output behavior of the expert selector: instead of a single neuron activating, the analysis investigates which expert module each token is routed to. Turner et al. (2023) propose steering middle-layer activations to improve model outputs. Voita et al. (2023) uses OPT model to do analysis for FFM neurons. Luo et al. (2025), using activations to understand the semantic information in LLMs. However, most papers analyze activations using activation scores, hidden states, or broader definitions of activation. Few works directly examine the activations right after the non-linear activation functions in the MLP.

## C    LIMITATIONS

Different meaningless tokens lead to varying performance outcomes. We only know that this difference arises from their differing degrees of transformation, but the underlying reason why different tokens cause such phenomena remains unclear. Meanwhile, we assume that meaningless tokens can be identified by LLMs in the first layer. Therefore, in our analysis, we focus only on their impact on meaningful tokens and how this interaction influences model performance. As such, we ignore the meaningless tokens themselves. Future work can further investigate the results when explicitly considering meaningless tokens. We restrict our analysis to the first layer, as it is the only layer where the attention scores exhibit a clear phenomenon (see Figure 2). Future work may extend this investigation to examine whether similar effects arise in deeper layers.

## D    TIME COMPLEXITY

In this section, we will analyze the time complexity of our method in the MLP. In the first layer's MLP, we have batch size $B$, sequence length $S$, feed forward dimensions $D_f$, model dimension $D_{model}$. For MLP, the module of time complexity contains gate project, up project and down project. The time complexity of each module is $O(2BSD_fD_{model})$, thus the total of MLP is:

$$\mathbf{T}_{mlp} = O(BSD_fD_{model}), \tag{11}$$

For ARM module, the operation contains: calculating MAD, comparing threshold, calculating proportion $p$, selecting elements that need to be changed. The time complexity of all above operations is $O(BSD_f)$. So the time complexity of ARM is:

$$\mathbf{T}_{ARM} = O(BSD_f), \tag{12}$$

The comparison between the time complexity of ARM and MLP is $\frac{1}{D_{model}}$. When $D_{model}$ equals to 4096. This proportion value is approximately $\frac{1}{2*4096} \approx 1.2 \times 10^{-4}$ at the level of one ten-thousandth. Therefore, we believe that the time complexity of ARM can be disregarded in MLP layer.

## E    PROOF

### E.1    SCALING FACTOR CAUSE VARIANCE RISE

**Lemma E.1.** *In LLMs, RMSNorm uses $\varepsilon > 0$; hence $J_q(x_0)$ is bounded and $\|x_0\| \geq \varepsilon$*

For every $\lambda$, we have:

$$x(\lambda) = r + \lambda UA, y(\lambda) = \text{RMS}(x(\lambda)), z_j(\lambda) = w_j^\top y(\lambda), \tag{13}$$

For every $\Delta\lambda$, we have:

$$z_j(\lambda + \Delta\lambda) \approx z_j(\lambda) + g_j(\lambda)\Delta\lambda, g_j(\lambda) = w_j^\top J_q(x(\lambda))UA, \tag{14}$$

For $\text{Var}_j$ we have following proof:

$$\Delta\text{Var}_j \triangleq \text{Var}\big[z_j(\lambda + \Delta\lambda)\big] - \text{Var}\big[z_j(\lambda)\big]$$
$$\approx 2\text{Cov}\big(z_j(\lambda), g_j(\lambda)\big)\Delta\lambda + \text{Var}\big[g_j(\lambda)\big](\Delta\lambda)^2.$$
$$\Delta\text{Var}_j \geq -2\big|\text{Cov}(z_j, g_j)\big||\Delta\lambda| + \text{Var}(g_j)(\Delta\lambda)^2.$$
$$|\Delta\lambda| > \frac{2\big|\text{Cov}(z_j, g_j)\big|}{\text{Var}(g_j)}$$

Meanwhile, we also need to have:

$$\Delta\text{Var}_j \geq -2\big|\text{Cov}(z_j, g_j)\big||\Delta\lambda| + A(\Delta\lambda)^2 - \frac{K}{6}|\Delta\lambda|^3. \tag{15}$$

K is upper bound of $\text{Var}[z_j(\lambda)]$, thus we have a range:

$$\frac{2\big|\text{Cov}(z_j, g_j)\big|}{\text{Var}(g_j)} \leq \Delta\lambda \leq \frac{3\text{Var}(g_j)}{\text{K}}. \tag{16}$$

For every $|\Delta\lambda|$, if it is in this range, we will have $\Delta\text{Var}_j > 0$. Specially, when $|\Delta\lambda|$ becomes larger, the quadratic term dominates, and A increases monotonically and eventually becomes positive.

### E.2 BIAS FACTOR CAUSE VARIANCE RISE

**Lemma E.2.** *The bias we add is a uniform distribution sampled independently each time and does not depend on the specific value of attention output.*

**Lemma E.3.** *In LLM's high dimensions, bias has a nonzero tangential component and* $w_j^\top J_q(x_0)W \neq 0$.

According to above lemmas we have:

$$\mathrm{Var}[z_j] \approx e_j^\top W_{gate}J_q(x_0)W\Sigma_\sigma W^\top J_q(x_0)^\top W_{gate}^\top e_j \tag{17}$$

Thus, we have $\Delta V_j > 0$.

## F MORE ANALYSIS

### F.1 THE IMPACT OF INSERTING LENGTH OF MEANINGLESS TOKENS

In this section, we analyze the relationship between the length of inserted tokens and the performance of LLMs. We evaluate five models on MATH-500 while varying the number of inserted tokens from 0 to 70. The results are shown in Figure 8. We observe that when the inserted sequence is relatively short, the models outperform the baseline, although their accuracy fluctuates. However, when too many tokens are inserted, performance drops sharply. This occurs because, as the length of the inserted tokens increases, their influence on the attention output values accumulates (as shown in Equation 1). Once this accumulation reaches a critical level, it no longer produces a small, benign effect; instead, it alters the model's internal semantic structure and degrades its performance.

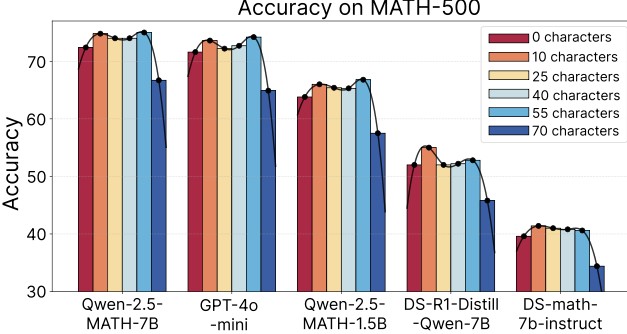

Figure 8: The relationship between the length of inserting tokens and the performance of models.

### F.2 THE IMPACT OF INSERTING POSITION OF MEANINGLESS TOKENS.

In the previous section, we demonstrated that inserting meaningless tokens between the system prompt and the question leads to improved model performance. In this section, we further investigate the effect of inserting meaningless tokens at different positions. Specifically, we consider four settings: ❶ the beginning of the system prompt, ❷ between the system prompt and the question, ❸ the end of the input, and ❹ a random position within the input. The results are reported in Table 4. We observe that only inserting tokens between the system prompt and the question yields performance gains. In contrast, appending tokens to the end of the input causes the model to simply repeat them, leading to zero accuracy. Inserting tokens at random positions disrupts the original semantic structure of the sentence, while inserting them at the beginning alters the values of the system prompt itself, introducing extra terms as shown in Equation 1. We hypothesize that this disrupts the intended initialization and interferes with the task the model is expected to process. Therefore, the most effective position for inserting meaningless tokens is between the system prompt and the question.

| Model | Math-500 (Pass@1) | AIME 2022-2024 (Pass@1) |
|---|---|---|
| w/o meaningless tokens | 72.4 | 23.1 |
| position ❶ | 69.6 | 21.1 |
| position ❷ | 75.0 | 23.3 |
| position ❸ | 0.0 | 0.0 |
| position ❹ | 51.2 | 21.1 |

Table 4: Performance on Math-500 and AIME 2022-2024 after inserting meaningless tokens in different positions.

### F.3 THE IMPACT OF INSERTING TYPE OF MEANINGLESS TOKENS

In this section, we examine the influence of inserting different types of meaningless tokens on reasoning tasks. In our experiments, we insert varying lengths of slashes ("/") and question marks ("?") into the inputs and select the best-performing configuration from each set. As shown in Table 5, different types of meaningless tokens produce varying impacts on LLM performance, and no single unified pattern emerges. We attribute this to the fact that different token types carry distinct representational values, leading to different effects of attention during the transformation. Moreover, the sensitivity of individual questions to such transformations also varies. Consequently, the impact of meaningless tokens differs across tasks and models.

Table 5: Accuracy of LLM on two mathematical reasoning datasets with inserting different kinds of meaningless tokens.

| Methods | MATH-500 | | | AIME 2022-2024 | | |
|---|---|---|---|---|---|---|
| | w/o Mless | \ | ? | w/o Mless | \ | ? |
| Qwen2.5-Math-1.5b | 63.6 | **66.8** | 58.2 | 14.4 | **18.8** | 16.1 |
| Qwen2.5-Math-7b | 72.4 | **75.0** | 69.6 | 23.3 | **24.4** | 22.2 |
| DeepSeek-R1-Distill-Qwen-7B | 52.0 | **55.0** | 53.6 | 3.3 | 3.3 | **4.4** |
| DeepSeek-Math-7b-instruct | 39.6 | 41.4 | **43.4** | 7.8 | 12.2 | **12.5** |
| Llama-3.1-8B-Instruct | 35.4 | **36.6** | 34.2 | 11.1 | 7.8 | **13.3** |
| Qwen-2.5-32B-Instruct | 80.8 | 81.0 | **81.6** | 18.9 | 20.0 | **21.1** |

### F.4 WHY WE ONLY ANALYZE FIRST LAYER

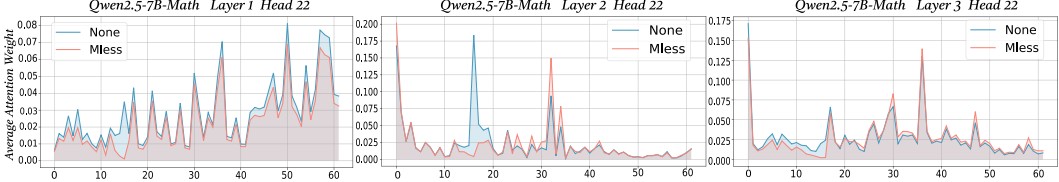

Figure 9: Average attention weights for later tokens in Layers 1 to 3 of Qwen2.5-7B-Math.

In this section, we explain why our analysis and redistribution of activations focus exclusively on the first layer. As shown in Figure 9 and Figure 10 , we present the average attention weights of later tokens in Layers 1, 2, and 3 of Qwen2.5-7B-Math. We observe that only the first layer exhibits a clear and consistent phenomenon: after inserting meaningless tokens, the average attention weights decrease to a noticeable extent, suggesting that meaningless tokens directly alter the initial allocation of attention. In contrast, Layers 2 and 3 do not display such regularity—the average attention weights with and without meaningless tokens show no systematic relationship. In the layers beyond layer 3, the phenomenon is even less obvious. Consequently, later layers do not undergo an affine transformation of this type. We hypothesize that this disappearance of the phenomenon arises because, beyond the first layer, the model has already integrated and mixed substantial semantic information through residual connections. From the second layer onward, the model begins to reconstruct and redistribute information, thereby diminishing the direct effect of meaningless tokens on average attention weights. In other words, the role of meaningless tokens becomes less distinguishable once meaningful contextual representations dominate, which explains why the first layer is the most critical point for observing and leveraging this effect.

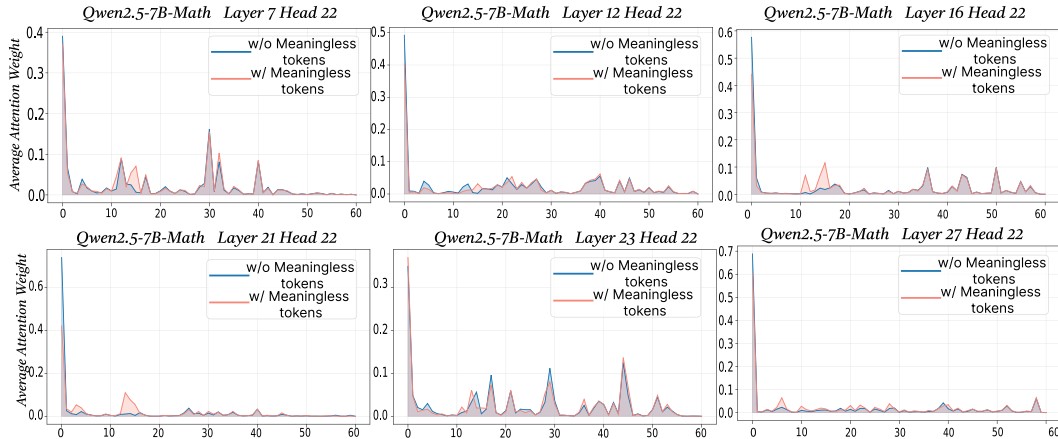

Figure 10: The average attention weights on the layers beyond layer. We can find that the. After adding meaningless tokens, subsequent layers do not show the same level of attention to meaningful words as the first layer after processing.

## F.5 REPEAT MEANINGFUL TOKENS' EFFECTIVENESS

In this section, we investigate whether adding meaningful tokens can play a role similar to meaningless tokens. Specifically, we insert a long sequence of repeated tokens that are semantically irrelevant to the question. For example, we add 55 repetitions of "he" between the system prompt and the question. The results, shown on the left of Figure 11, indicate that even such repeated but irrelevant meaningful tokens lead to an improvement in model performance. To better understand this effect, we further visualize the average attention weights after inserting these tokens, as presented on the right of Figure 11. The results reveal that the activation changes induced by repeated meaningful tokens closely resemble those caused by meaningless tokens, and the inserted tokens receive similar attention patterns which means the weight value of inserted part's are similar. Taken together, these findings suggest that when repeated tokens are inserted at appropriate positions without introducing additional semantic content, LLMs are able to recognize them as irrelevant. Consequently, they trigger a redistribution of activations in the MLP, ultimately improving model performance.

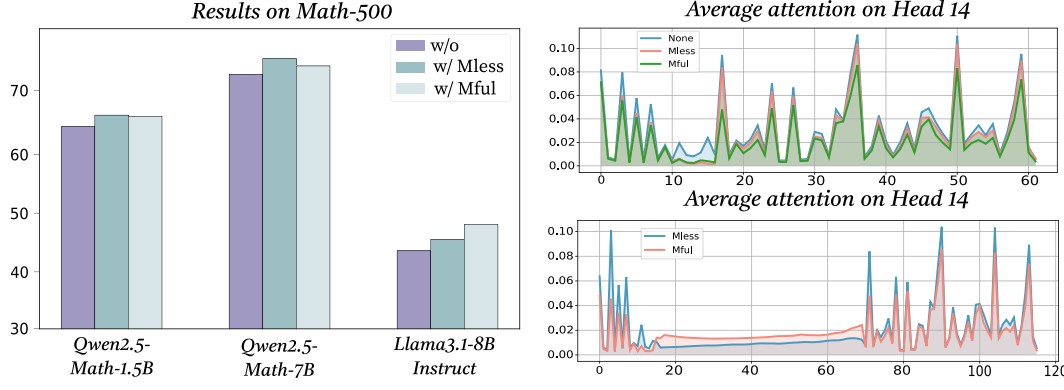

Figure 11: The left panel illustrates a comparison between adding repeated meaningful tokens and meaningless tokens, while the right panel presents the average attention weights resulting from the addition of meaningful and meaningless tokens.

## F.6 WHY RANDOM SENTENCE IS USELESS

When additional tokens are inserted into a sentence, both the attention weights and the resulting attention outputs exhibit consistent patterns: the weights assigned to the original tokens decrease,

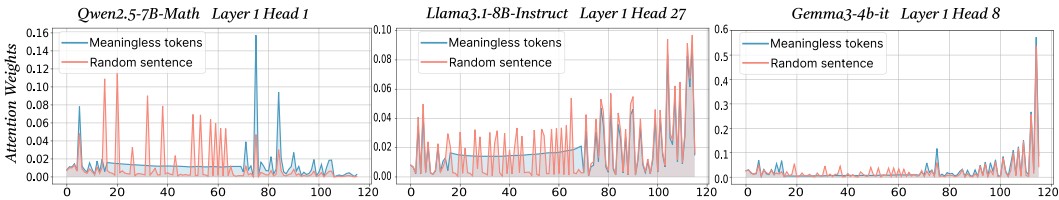

Figure 12: The average attention weights of adding meaningless tokens and random sentence.

while the attention outputs gain additional values from the inserted tokens. In this section, we analyze why adding repeated tokens can enhance the performance of LLMs, whereas inserting random or unrelated sentences can have a detrimental effect. The results are shown in Figure 12. We observe that the attention weights associated with the random sentence are highly diverse, and their corresponding value vectors also differ substantially. In contrast, the repeated meaningless tokens exhibit more uniform attention weights and nearly identical value vectors. Consequently, compared with repeated meaningless tokens, a random sentence introduces not only numerical fluctuations but also a pronounced directional shift in the attention outputs—one that carries additional semantic information. The formula of RMSNorm is:

$$\text{RMSNorm}(x) = \gamma \odot \frac{x}{\sqrt{\frac{1}{d} \sum_{i=1}^{d} x_i^2 + \epsilon}}, \tag{18}$$

where $\gamma$ is a learnable rescaling vector and $\epsilon$ ensures numerical stability. For repeated meaningless tokens, the effect manifests as a small and uniform directional bias on the input to RMSNorm, producing only a minor numerical perturbation in its output. In contrast, inserting a random sentence introduces high-rank and structured semantic signals that RMSNorm cannot simply absorb. This leads to systematic shifts in the output direction and subspace, thereby altering the model's internal semantic representations.

### F.7 THE OPTIMAL HYPERPARAMETER RANGE

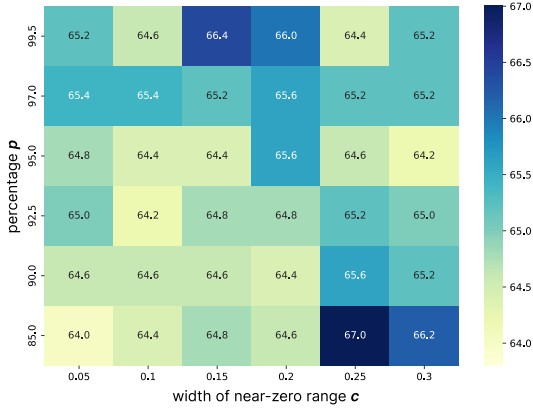

Figure 13: This figure illustrates how accuracy varies with changes in the parameters $p$ and $c$.

In this section, we investigate how the hyper-parameters—the percentage $\boldsymbol{p}$ and the width of the near-zero range $\boldsymbol{c}$—influence model performance on Math-500 when using Qwen2.5-1.5B-Math. The results are summarized in Figure 13. As the figure illustrates, the accuracy does not change monotonically with either $p$ or $c$; instead, the best-performing settings emerge only within specific regions of the parameter space. This indicates that the choice of hyper-parameters is not trivial and cannot be reduced to cherry-picking. More concretely, we find that balanced combinations of $p$ and $c$ lead to more stable improvements. High accuracy is typically concentrated in two regions: when $p$ is large and $c$ is small, or conversely, when $c$ is large and $p$ is small. In these cases, the redistribution

Table 6: Complete results of several models on non-reasoning tasks.

| Model | Setting | GSM8K | ARC-E | ARC-C | MMLU | BoolQ | HellaSwag | OpenBookQA |
|---|---|---|---|---|---|---|---|---|
| Qwen2.5 Math-1.5B | Baseline | 78.0 | 39.3 | 35.0 | 32.1 | 32.6 | 39.1 | 42.0 |
| | ARM | 78.6 | 39.3 | 35.4 | 32.1 | 33.4 | 39.5 | 42.4 |
| | Improve Rate (%) | **0.6↑** | 0 | **0.4↑** | 0 | **0.8↑** | **0.4↑** | **0.4↑** |
| Qwen2.5 Math-7B | Baseline | 83.8 | 49.7 | 47.9 | 36.9 | 38.6 | 46.9 | 47.6 |
| | ARM | 83.8 | 49.7 | 47.0 | 37.5 | 38.7 | 47.1 | 47.9 |
| | Improve Rate (%) | 0 | 0 | 0 | **0.6↑** | **0.1↑** | **0.2↑** | **0.3↑** |
| Llama3.1 8B-Instruct | Baseline | 80.0 | 46.6 | 49.0 | 38.6 | 43.3 | 56.8 | 52.8 |
| | ARM | 82.4 | 47.1 | 48.7 | 38.2 | 43.2 | 57.3 | 50.8 |
| | Improve Rate (%) | **2.4↑** | **0.5↑** | **-0.3↓** | **-0.4↓** | **-0.1↓** | **0.5↑** | **-2.0↓** |
| Gemma3 4b-it | Baseline | 86.8 | 47.1 | 44.5 | 33.9 | 45.0 | 42.0 | 41.0 |
| | ARM | 86.8 | 47.1 | 45.0 | 34.1 | 45.2 | 42.0 | 42.0 |
| | Improve Rate (%) | 0 | 0 | **0.5↑** | **0.2↑** | **0.2↑** | 0 | **1.0↑** |

mechanism introduced by ARM effectively amplifies informative activations while suppressing uninformative near-zero activations. Outside of these regions, however, the performance of the model degrades, suggesting that poorly chosen hyper-parameters may distort the activation distribution rather than enhance it. These observations highlight the importance of aligning hyper-parameter choices with the intrinsic properties of activation distributions. To maximize the benefits of ARM, one must take into account both the proportion of near-zero activations and the magnitude of the maximum activation values, thereby ensuring that $p$ and $c$ are set within an appropriate interval. By doing so, ARM can operate in its most effective regime, consistently improving model reasoning performance rather than introducing instability. From each row and column, we can see that the performance of LLMs after ARM depends on both $p$ and $c$. Since they are equally important, the optimal performance is determined by the range of these two parameters.

## G    MORE EXPERIMENTS

### G.1    RESULTS ON NON-REASONING TASKS

In this section, we present supplementary results on non-reasoning benchmarks, including ARC-C, MMLU, BoolQ, and OpenBookQA, as shown in Table 6. Across all evaluated models, the application of our method yields only marginal variations in performance. For most models and tasks, it either produces slight improvements or maintains parity with the baseline (i.e., vanilla model performance without any inference-time trick), suggesting that the redistribution of activations has little impact when the task primarily requires factual recall or pattern recognition rather than multi-step reasoning. A minor performance drop is observed only on a small subset of tasks with Llama-3.1-8B-Instruct, which we attribute to model-specific characteristics or sensitivity to activation perturbations. These findings indicate that our approach exerts negligible influence on non-reasoning tasks and, in most cases, does not introduce adverse effects on task accuracy. This observation further supports our central claim: the benefits of activation redistribution are **most pronounced in reasoning-oriented scenarios**, while **in non-reasoning settings the method remains stable and does not compromise the model's inherent ability** to answer factual or knowledge-intensive questions.

### G.2    RESULTS ON BASE MODEL

In this section, we evaluate the effect of applying ARM to base models and report their performance on Math-500 and AIME 2022-2024 using Qwen2.5-1.5B, Qwen2.5-7B, and Qwen2.5-32B. Since these models achieve accuracy above 5%, we consider them capable of tackling these tasks. In contrast, models such as Llama3.1-8B and Gemma3-4B-PT exhibit poor performance and are therefore excluded from the evaluation. The results in Table 7 show that incorporating ARM into the MLP layers of base models yields measurable performance gains on reasoning tasks, although the improvements are generally smaller than those observed for reasoning, oriented models. We attribute this gap to the weaker inherent reasoning abilities of base models. While activation redistribution can still enhance their internal representations, it may not strongly affect how they process key numerical

Table 7: Performance on Math-500 and AIME 2022-2024 after incorporating ARM into the MLP in non-reasoning model.

| Model | Setting | Math-500 (Pass@1) | AIME 2022-2024 (Pass@1) |
|---|---|---|---|
| Qwen2.5-1.5B | Baseline | 67.8 | 14.4 |
| | ARM | 68.2 | 14.4 |
| | **Improve Rate (%)** | **0.4↑** | 0 |
| Qwen2.5-7B | Baseline | 50.4 | 15.6 |
| | ARM | 50.6 | 16.7 |
| | **Improve Rate (%)** | **0.2↑** | **1.1↑** |
| Qwen2.5-32B | Baseline | 77.2 | 27.8 |
| | ARM | 77.4 | 28.9 |
| | **Improve Rate (%)** | **0.2↑** | **1.1↑** |

or symbolic elements, such as digits and operators, compared with models trained specifically for reasoning.

Table 8: Performance on Math-500 and AIME 2022-2024 after incorporating ARM into the MLP.

| Model | Setting | Math-500 (Pass@1) | AIME 2022-2024 (Pass@1) |
|---|---|---|---|
| Qwen2.5 Math-1.5B | Baseline | 63.8 | 14.4 |
| | ARM | 67.8 | **18.9** |
| | Best-of-N(N=5) | 69.4 | 14.4 |
| | Best-of-N + ARM | **71.2** | **18.9** |
| Qwen2.5 Math-7B | Baseline | 72.4 | 23.3 |
| | ARM | **73.4** | **25.6** |
| | Best-of-N(N=5) | 72.8 | 23.3 |
| | Best-of-N + ARM | **73.4** | **25.6** |

## G.3 INFERENCE TIME TRICK COMPARISON

To more comprehensively evaluate the robustness, effectiveness, and compatibility of ARM with established inference-time scaling techniques, we further compare its performance against the widely used Best-of-N sampling approach during inference. Specifically, Table 8 summarizes the results obtained by applying ARM alone, Best-of-N sampling alone, and their combined usage on two representative reasoning benchmarks. For all settings, we fix the generation hyperparameters to a temperature of 0.5 and a top_p of 0.95 to ensure a consistent sampling regime. As demonstrated in the table, both ARM and Best-of-N independently yield improvements over the baseline, and their combination produces an even larger performance gain, suggesting that ARM complements rather than competes with existing inference-time strategies. These findings collectively underscore the practical value and scalability of ARM as a lightweight inference-time method for enhancing reasoning capabilities across diverse tasks.

Table 9: The hyper parameters in 7 models on three benchmarks. For Qwen and Llama, we using near-zero range $c$ to choose proportion, so $p$ is dash("-"). But for Gemma, due to the activation distribution, we directly decide to skip setting $c$ and choose $p$. So here, $c$ is dash("-"). If the task performance doesn't improve, we replace hyper-parameters with dash("-").

| Model | GPQA Diamond | | Math-500 | | AIME 2022-2024 | |
|---|---|---|---|---|---|---|
| | $c$ / $p$ | $p_1$ | $c$ / $p$ | $p_1$ | $c$ / $p$ | $p_1$ |
| Qwen2.5-Math-1.5B | 0.15/- | 99.5 | 0.13/- | 99.5 | 0.13/- | 99.5 |
| Qwen2.5-Math-7B | 0.2/- | 99.5 | 0.1/- | 95.0 | 0.05/- | 90.0 |
| Qwen2.5-7B-Instruct | 0.15/- | 99.5 | 0.1/- | 99.5 | - | - |
| Qwen2.5-32B-Instruct | 0.05/- | 90.5 | - | - | 0.13/- | 99.5 |
| Llama3.1-8B-Instruct | 0.45/- | 80.0 | 0.32/- | 90.0 | 0.32/- | 90.0 |
| Gemma3-4B-it | -/0.5 | 96.5 | -/0.25 | 85.0 | -/0.25 | 96.5 |
| Gemma3-27B-it | -/0.5 | 96.5 | -/0.25 | 85.0 | -/0.25 | 85.0 |

## H    THE EXAMPLE OF HYPER PARAMETERS

The selection of $p_1$ and $c$ depends on the distribution of model activations after the activation function. In most cases, when the frequency of near-zero activations greatly exceeds that of other values (as in Qwen), the value of $c$ should be chosen smaller. In contrast, for models like LLaMA, $c$ should be chosen larger. When the proportion of near-zero activations is extremely high (as in Gemma), we recommend directly setting the modification proportion to a value larger than $p_{\max}$. For the choice of $p_1$, we generally advise selecting more than $80\%$, which covers regions with higher activation frequencies and thus exerts a stronger influence on near-zero values. Because the proportion of extremely high-frequency activations is small, this choice will not cause large deviations in the near-zero values. Overall, although models from the same family may behave differently across tasks and parameter scales, the selection ranges of $p_1$ and $c$ remain relatively consistent within each family. Some examples are in Table 9.

## I    MORE AVERAGE ACTIVATION WEIGHTS

In this section, we present additional figures of average attention weights to further validate the previously observed phenomenon. When computing the average attention weights for Llama3.1 and Gemma3, we exclude the first token to make the effect clearer, as position tokens tend to absorb most of the attention. The results are in Figure 14, Figure 15, Figure 16, Figure 17, Figure 18, Figure 19, Figure 20, **??**. From results, we can see that in the most layers, models will have same phenomenon like we mentioned before.

## J    EXAMPLES OF MODEL'S OUTPUT CHANGE OF ADDING MEANINGLESS

In this section, we show some examples where, after adding meaningless tokens, the model can turn wrong answers into correct ones. The specific examples are provided in Appendix J.

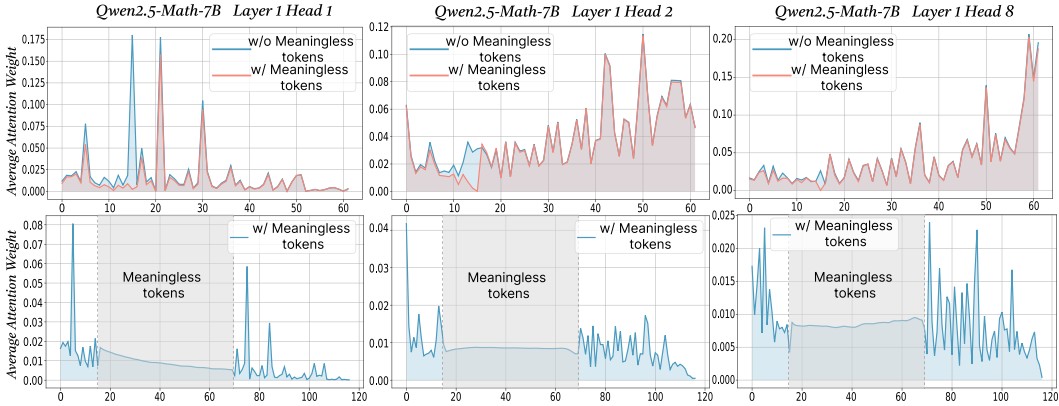

Figure 14: The average attention weights of Qwen2.5-Math-7B in Head 1, 2, 8.

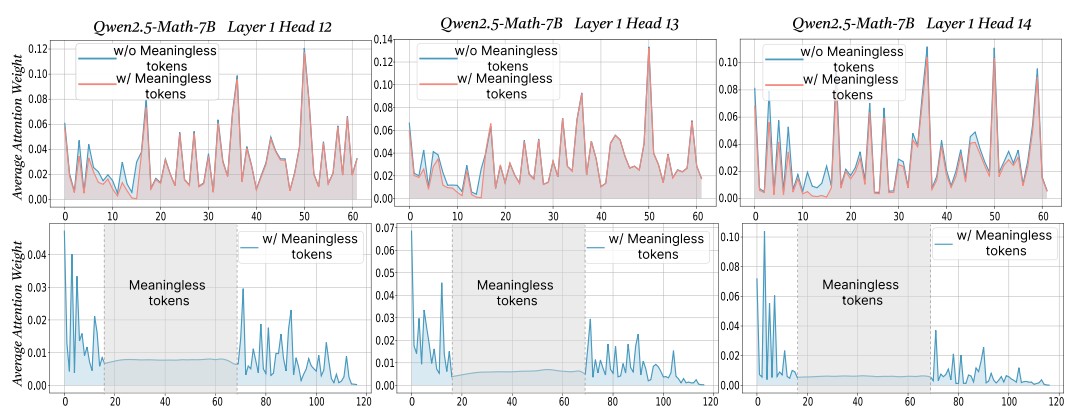

Figure 15: The average attention weights of Qwen2.5-Math-7B in Head 12, 13, 14.

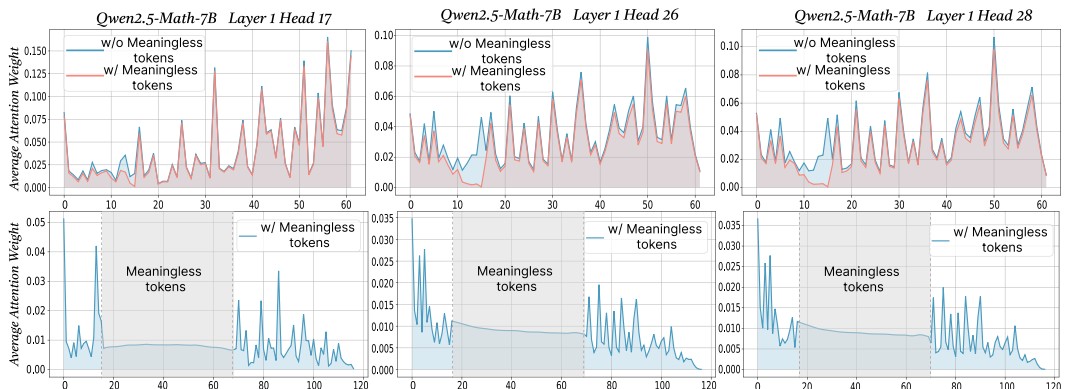

Figure 16: The average attention weights of Qwen2.5-Math-7B in Head 17, 26, 28.

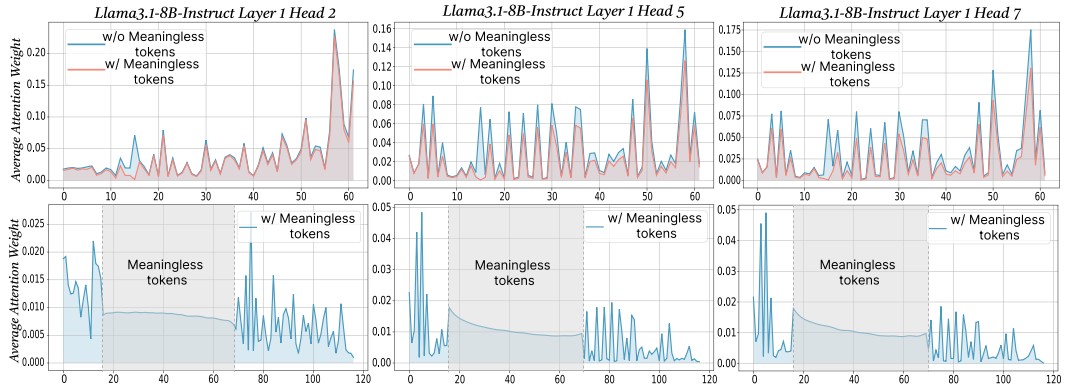

Figure 17: The average attention weights of Llama3.1-8B-Instruct in Head 2, 5, 7.

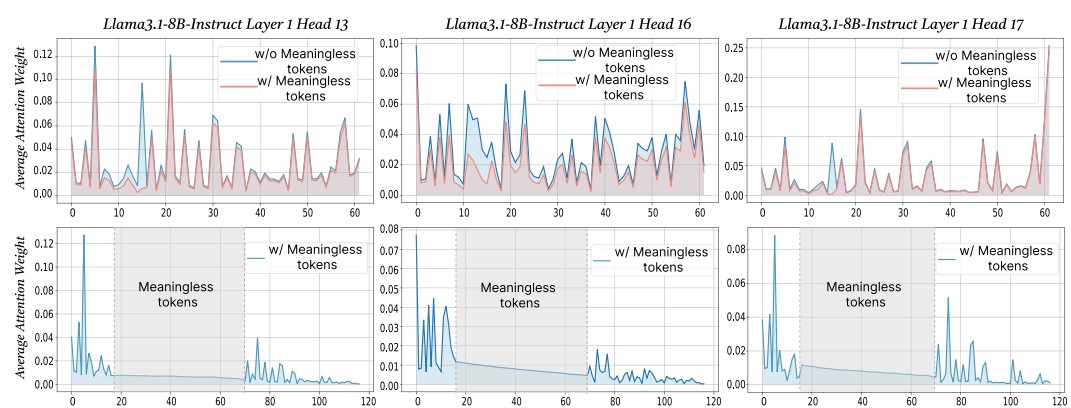

Figure 18: The average attention weights of Llama3.1-8B-Instruct in Head 13, 16, 17.

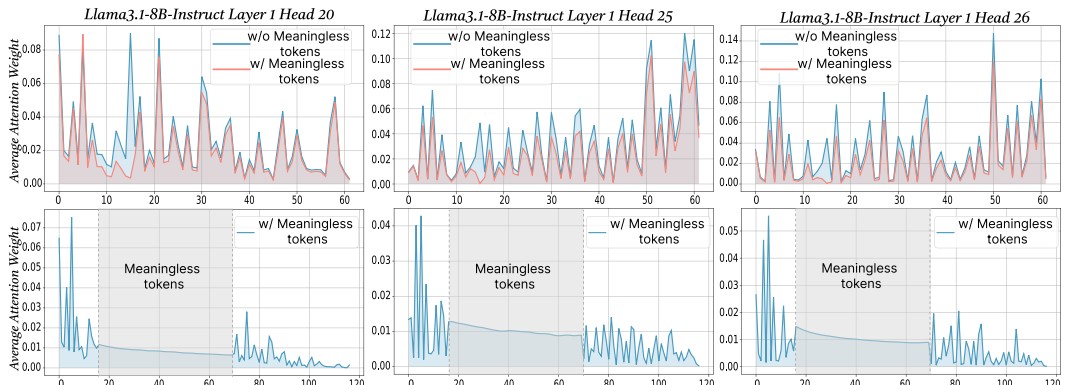

Figure 19: The average attention weights of Llama3.1-8B-Instruct in Head 20, 25, 26.

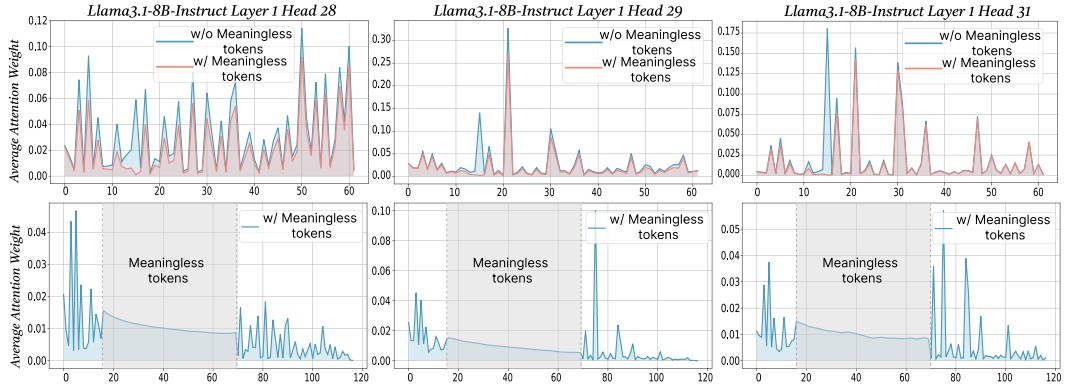

Figure 20: The average attention weights of Llama3.1-8B-Instruct in Head 28, 29, 31.

—————————————————————— - **Question w/o Meaningless Tokens** ——————————————————————

**System Prompt:** You are an expert mathematician. Solve the following problem carefully. Put your final answer within a {}.

**User Prompt:** Below is the graph of $y = a\sin(bx + c) + d$ for some positive constants $a$, $b$, $c$, and $d$. Find the smallest possible value of $c$.

[asy]import TrigMacros;

size(400);

real f(real x) return 2*sin(3*x + pi) + 1;

draw(graph(f,-3*pi,3*pi,n=700,join=operator ..),red);

trig_axes(-3*pi,3*pi,-4,4,pi/2,1);

layer();

rm_trig_labels(-5,5, 2);

label("1", (0,1), E);

label("2", (0,2), E);

label("3", (0,3), E);

label("−1", (0,-1), E);

label("−2", (0,-2), E);

label("−3", (0,-3), E); [/asy]

—————————————————————— **Answer w/o Meaningless tokens** ——————————————————————-

To solve the problem, we need to analyze the given graph of the function $y = a\sin(bx + c) + d$ and determine the smallest possible value of $c$.

Step-by-Step Analysis:

1. **Identify the Amplitude $a$:** The amplitude $a$ is the maximum value of the function minus the minimum value of the function divided by 2. From the graph, the maximum value is 3 and the minimum value is -1. Therefore, the amplitude $a$ is: a $= \frac{3-(-1)}{2} = \frac{4}{2} = 2$

2. **Identify the Vertical Shift $d$:** The vertical shift $d$ is the average of the maximum and minimum values of the function. Therefore, the vertical shift $d$ is:$d = \frac{3+(-1)}{2} = \frac{2}{2} = 1$

3. **Identify the Period and Frequency $b$:** The period of the sine function is the distance between two consecutive peaks or troughs. From the graph, the period appears to be $\frac{2\pi}{3}$. The period of the sine function $\sin(bx + c)$ is $\frac{2\pi}{b}$. Therefore, we can solve for $b$:$\frac{2\pi}{b} = \frac{2\pi}{3} \implies b = 3$

4. **Identify the Phase Shift $c$:** The phase shift $c$ can be determined by looking at the horizontal shift of the graph. The standard sine function $\sin(x)$ starts at 0 when $x = 0$. However, the given function starts at its maximum value when $x = 0$. This means the phase shift $c$ must be such that $\sin(c) = 1$. The smallest positive value of $c$ that satisfies this is $c = \frac{\pi}{2}$.

Therefore, the smallest possible value of $c$ is $\frac{\pi}{2}$.

—————————————————————— - **Question w/ Meaningless Tokens** ——————————————————————

**System Prompt:** You are an expert mathematician. Solve the following problem carefully. Put your final answer within a {}.

**User Prompt:** \\\\\\\\\\\\\\\\\\\\\\\\\\\\\\\\\\\\\\\\\\\\\\\\\\\\\\\\\\\\\\\\\\Below is the graph of $y = a\sin(bx + c) + d$ for some positive constants $a$, $b$, $c$, and $d$. Find the smallest possible value of $c$.

[asy]import TrigMacros;

size(400);

real f(real x) return 2*sin(3*x + pi) + 1;

draw(graph(f,-3*pi,3*pi,n=700,join=operator ..),red);

trig_axes(-3*pi,3*pi,-4,4,pi/2,1);

layer();

rm_trig_labels(-5,5, 2);

label("1", (0,1), E);

label("2", (0,2), E);

label("3", (0,3), E);

label("−1", (0,-1), E);

label("−2", (0,-2), E);

label("−3", (0,-3), E); [/asy]

——————————————————————-**Answer w/ Meaningless tokens**——————————————————————-

Step-by-Step Analysis:

1. **Identify the Characteristics of the Graph:** - The graph oscillates between 3 and -1, indicating the amplitude $a$ and the vertical shift $d$. - The period of the sine wave is $\frac{2\pi}{b}$. - The phase shift is given by $-\frac{c}{b}$.

2. **Determine the Amplitude and Vertical Shift:** - The amplitude $a$ is half the distance between the maximum and minimum values of the function. Here, the maximum value is 3 and the minimum value is -1, so the amplitude $a = \frac{3-(-1)}{2} = 2$. - The vertical shift $d$ is the average of the maximum and minimum values. Here, $d = \frac{3+(-1)}{2} = 1$.

3. **Determine the Period:** - From the graph, we can see that the period is $\frac{2\pi}{3}$. This means $b = 3$.

4. **Determine the Phase Shift:** - The phase shift is the horizontal shift of the sine wave. From the graph, we can see that the sine wave starts at its minimum value at $x = 0$. This means the phase shift is $\pi$ to the left, so $-\frac{c}{b} = -\frac{\pi}{3}$. Therefore, $c = \pi$.

Conclusion:

The smallest possible value of $c$ is $\pi$.

─────────────────────── **Question w/ Meaningless Tokens** ───────────────────────

**System Prompt:** You are an expert mathematician. Solve the following problem carefully. Put your final answer within a {}.

**User Prompt:** Find the greatest integer less than $(\sqrt{7} + \sqrt{5})^6$. (Do not use a calculator!)

─────────────────────── **Answer w/o Meaningless tokens** ───────────────────────

To find the greatest integer less than $(\sqrt{7} + \sqrt{5})^6$, we can use the binomial theorem and some algebraic manipulation. Let's denote $a = \sqrt{7} + \sqrt{5}$ and $b = \sqrt{7} - \sqrt{5}$. Notice that $a$ and $b$ are conjugates, and their product is $ab = (\sqrt{7} + \sqrt{5})(\sqrt{7} - \sqrt{5}) = 7 - 5 = 2$.

We can express $a^6$ in terms of $b^6$ using the binomial theorem. Since $b = \sqrt{7} - \sqrt{5}$ is a small positive number (approximately 0.196), $b^6$ will be a very small positive number (much less than 1). Therefore, $a^6 + b^6$ will be an integer because all the irrational terms will cancel out when we add $a^6$ and $b^6$.

Let's calculate $a^6 + b^6$: $a^6 + b^6 = (\sqrt{7} + \sqrt{5})^6 + (\sqrt{7} - \sqrt{5})^6$

Using the binomial theorem, we can expand both terms and notice that the irrational terms will cancel out when we add them together. We can use Python to compute this accurately.

First, let's calculate $a^6$ and $b^6$ directly using Python, and then find the integer part of $a^6$. '''python import math

#Define the values a = math.sqrt(7) + math.sqrt(5) b = math.sqrt(7) - math.sqrt(5)

#Calculate $a^6$ and $b^6$ a_6 = a**6 b_6 = b**6

#Calculate the sum $a^6 + b^6$ sum_a_b_6 = a_6 + b_6

#The greatest integer less than $a^6$ is the integer part of $b^6$ greatest_integer_less_than_a_6 = math.floor(a_6)

print((a_6, b_6, sum_a_b_6, greatest_integer_less_than_a_6))

# output

(10396.000015740316, 0.000015740316569499255, 10396.000031480633, 10396)

The calculations show that:

$$a^6 \approx 10396.000015740316$$

$$b^6 \approx 0.000015740316569499255$$

$$a^6 + b^6 \approx 10396.000031480633$$

Since $b^6$ is a very small number, $a^6$ is very close to the integer 10396.

Therefore, the greatest integer less than $a^6$ is: 10395

─────────────────────── **Question w/ Meaningless Tokens** ───────────────────────

**System Prompt:** You are an expert mathematician. Solve the following problem carefully. Put your final answer within a {}.

**User Prompt:** \\\\\\\\\\\\\\\\\\\\\\\\\\\\\\\\\\\\\\\\\\\\\\\\\\\\\\\\\\\\Find the greatest integer less than $(\sqrt{7} + \sqrt{5})^6$. (Do not use a calculator!)

───────────────────────**Answer w/ Meaningless tokens**───────────────────────

To find the greatest integer less than $(\sqrt{7} + \sqrt{5})^6$, we will use a combination of algebraic manipulation and properties of binomial expansions. Let's denote $a = \sqrt{7}$ and $b = \sqrt{5}$. We need to find the greatest integer less than $(a + b)^6$.

First, consider the binomial expansion of $(a + b)^6$: $(a + b)^6 = \sum_{k=0}^{6} \binom{6}{k} a^{6-k} b^k$

Next, consider the binomial expansion of $(a - b)^6$:$(a - b)^6 = \sum_{k=0}^{6} \binom{6}{k} a^{6-k}(-b)^k = \sum_{k=0}^{6} \binom{6}{k} a^{6-k}(-1)^k b^k$

Now, add these two expansions:

$(a + b)^6 + (a - b)^6 = \sum_{k=0}^{6} \binom{6}{k} a^{6-k} b^k + \sum_{k=0}^{6} \binom{6}{k} a^{6-k}(-1)^k b^k$

Notice that terms where $k$ is odd will cancel out, and terms where $k$ is even will double:$(a + b)^6 + (a - b)^6 = 2\sum_{k \text{ even}} \binom{6}{k} a^{6-k} b^k$

The even values of $k$ are 0, 2, 4, and 6. Therefore:$(a+b)^6+(a-b)^6 = 2\left(\binom{6}{0}a^6 b^0 + \binom{6}{2}a^4 b^2 + \binom{6}{4}a^2 b^4 + \binom{6}{6}a^0 b^6\right)$

Calculate each term: $\binom{6}{0}a^6 b^0 = 1 \cdot (\sqrt{7})^6 \cdot 1 = 7^3 = 343$ $\binom{6}{2}a^4 b^2 = 15 \cdot (\sqrt{7})^4 \cdot (\sqrt{5})^2 = 15 \cdot 49 \cdot 5 = 3675$ $\binom{6}{4}a^2 b^4 = 15 \cdot (\sqrt{7})^2 \cdot (\sqrt{5})^4 = 15 \cdot 7 \cdot 25 = 2625$ $\binom{6}{6}a^0 b^6 = 1 \cdot 1 \cdot (\sqrt{5})^6 = 5^3 = 125$

Sum these terms: $343 + 3675 + 2625 + 125 = 6768$

Thus: $(a + b)^6 + (a - b)^6 = 2 \times 6768 = 13536$

Next, consider $(a - b)^6$. Since $a = \sqrt{7}$ and $b = \sqrt{5}$, we have: $a - b = \sqrt{7} - \sqrt{5}$

Since $\sqrt{7} > \sqrt{5}$, $a - b$ is a positive number less than 1. Therefore, $(a - b)^6$ is a very small positive number, specifically: $0 < (a - b)^6 < 1$

Thus: $(a + b)^6 = 13536 - (a - b)^6$

Since $(a - b)^6$ is very small, $(a + b)^6$ is slightly less than 13536.

Therefore, the greatest integer less than $(a + b)^6$ is: 13535.

———————————————— **Question w/ Meaningless Tokens** ————————————————

**System Prompt:** You are an expert mathematician. Solve the following problem carefully. Put your final answer within a {}.

**User Prompt:** Denali and Nate work for a dog walking business and are paid for each dog they walk. Denali is responsible for 16 dogs and Nate is responsible for 12 dogs. Under the company's new policy, they will be assigned or unassigned new dogs in groups of $x$ dogs. The ratio of Denali's pay to Nate's pay would be the same if Denali started walking $4x$ more dogs and Nate stayed at 12 dogs or if $x$ of Nate's dogs were reassigned to Denali. Find $x$ if $x \neq 0$.

———————————————— **Answer w/o Meaningless tokens** ————————————————

To solve this problem, we need to set up the equations based on the given conditions and solve for $x$.

1. **Initial Setup:** - Denali is responsible for 16 dogs. - Nate is responsible for 12 dogs. - The ratio of Denali's pay to Nate's pay is $\frac{16}{12} = \frac{4}{3}$.

2. **First Condition:** - If Denali starts walking $4x$ more dogs, she will be responsible for $16 + 4x$ dogs. - Nate stays at 12 dogs. - The new ratio of Denali's pay to Nate's pay is $\frac{16+4x}{12}$. - This ratio should be equal to $\frac{4}{3}$.

3. **Second Condition:** - If $x$ of Nate's dogs are reassigned to Denali, Denali will be responsible for $16 + x$ dogs and Nate will be responsible for $12 - x$ dogs. - The new ratio of Denali's pay to Nate's pay is $\frac{16+x}{12-x}$. - This ratio should also be equal to $\frac{4}{3}$.

We can set up the following equations based on the conditions: $\frac{16+4x}{12} = \frac{4}{3}$ $\frac{16+x}{12-x} = \frac{4}{3}$

Let's solve these equations step-by-step using Python and sympy.

```python
import sympy as sp
# Define the variable
x = sp.symbols('x')
# First equation: (16 + 4x) / 12 = 4 / 3
eq1 = sp.Eq((16 + 4*x) / 12, 4 / 3)
# Second equation: (16 + x) / (12 - x) = 4 / 3
eq2 = sp.Eq((16 + x) / (12 - x), 4 / 3)
# Solve the equations
solution1 = sp.solve(eq1, x) solution2 = sp.solve(eq2, x)
# Find the common solution
common_solution = set(solution1).intersection(set(solution2))
# Print the common solution
print(common_solution)
# output
2.00000000000000
```

The common solution to both equations is $x = 2$.

Therefore, the value of $x$ is 2.

———————————————— **Question w/ Meaningless Tokens** ————————————————

**System Prompt:** You are an expert mathematician. Solve the following problem carefully. Put your final answer within a {}.

**User Prompt:** \\\\\\\\\\\\\\\\\\\\\\\\\\\\\\\\\\\\\\\\\\\\\\\\\\\\\\\\\\\\\\\\\\\\\\\\\\\\Denali and Nate work for a dog walking business and are paid for each dog they walk. Denali is responsible for 16 dogs and Nate is responsible for 12 dogs. Under the company's new policy, they will be assigned or unassigned new dogs in groups of $x$ dogs. The ratio of Denali's pay to Nate's pay would be the same if Denali started walking $4x$ more dogs and Nate stayed at 12 dogs or if $x$ of Nate's dogs were reassigned to Denali. Find $x$ if $x \neq 0$.

————————————————**Answer w/ Meaningless tokens**————————————————

To solve the problem, we need to set up equations based on the given conditions and solve for $x$.

First, let's define the variables: - Let $p$ be the pay per dog. - Denali's current pay: $16p$ - Nate's current pay: $12p$

### Condition 1: Denali walks $4x$ more dogs
If Denali walks $4x$ more dogs, her new total number of dogs will be: $16 + 4x$ Her new pay will be: $(16 + 4x)p$ Nate's pay remains: $12p$

The ratio of Denali's new pay to Nate's pay is: $\frac{(16+4x)p}{12p} = \frac{16+4x}{12}$

### Condition 2: $x$ of Nate's dogs are reassigned to Denali
If $x$ of Nate's dogs are reassigned to Denali, Denali's new total number of dogs will be: $16 + x$ Nate's new total number of dogs will be: $12 - x$

The ratio of Denali's new pay to Nate's new pay is: $\frac{(16+x)p}{(12-x)p} = \frac{16+x}{12-x}$

### Setting the ratios equal
According to the problem, these two ratios are equal: $\frac{16+4x}{12} = \frac{16+x}{12-x}$

### Solving the equation
Cross-multiply to eliminate the fractions: $(16 + 4x)(12 - x) = 12(16 + x)$
Expand both sides: $192 + 48x - 16x - 4x^2 = 192 + 12x$ $192 + 32x - 4x^2 = 192 + 12x$
Subtract 192 from both sides: $32x - 4x^2 = 12x$
Subtract 12x from both sides: $20x - 4x^2 = 0$
Factor out 4x: $4x(5 - x) = 0$
Since $x \neq 0$, we solve: $5 - x = 0$ $x = 5$
Thus, the value of $x$ is: 5.

———————————————————— **Question w/o Meaningless Tokens** ————————————————————

**System Prompt:** You are an expert mathematician. Solve the following problem carefully. Put your final answer within a {}.

**User Prompt:** Consider the geometric sequence $\frac{125}{9}, \frac{25}{3}, 5, 3, \ldots$. What is the eighth term of the sequence? Express your answer as a common fraction.

———————————————————— **Answer w/o Meaningless tokens** ————————————————————

To find the eighth term of the geometric sequence, we need to determine the common ratio and then use the formula for the nth term of a geometric sequence.

The formula for the nth term of a geometric sequence is given by: $a_n = a_1 \cdot r^{n-1}$ where $a_n$ is the nth term, $a_1$ is the first term, $r$ is the common ratio, and $n$ is the term number.

First, we need to find the common ratio $r$. We can find $r$ by dividing the second term by the first term:

$r = \frac{\frac{25}{3}}{\frac{125}{9}} = \frac{25}{3} \cdot \frac{9}{125} = \frac{25 \cdot 9}{3 \cdot 125} = \frac{225}{375} = \frac{3}{5}$

Now that we have the common ratio $r = \frac{3}{5}$, we can find the eighth term $a_8$ using the formula for the nth term: $a_8 = a_1 \cdot r^{8-1} = \frac{125}{9} \cdot \left(\frac{3}{5}\right)^7$

Let's calculate this step by step: $\left(\frac{3}{5}\right)^7 = \frac{3^7}{5^7} = \frac{2187}{78125}$ $a_8 = \frac{125}{9} \cdot \frac{2187}{78125} = \frac{125 \cdot 2187}{9 \cdot 78125} = \frac{273375}{703125} = \frac{273375 \div 3375}{703125 \div 3375} = \frac{81}{2125}$

So the eighth term of the sequence is $\frac{81}{2125}$.

Let's confirm this with Python code. ```python # Define the first term and the common ratio
a1 = 125 / 9
r = 3 / 5
# Calculate the eighth term
n = 8
a8 = a1 * (r ** (n - 1))
# Convert the result to a fraction
from fractions import Fraction
a8_fraction = Fraction(a8).limit_denominator()
print(a8_fraction)
# output
81/2125

The eighth term of the geometric sequence is $\left\{\frac{81}{2125}\right\}$.

———————————————————— **Question w/ Meaningless Tokens** ————————————————————

**System Prompt:** You are an expert mathematician. Solve the following problem carefully. Put your final answer within a {}.

**User Prompt:** \\\\\\\\\\\\\\\\\\\\\\\\\\\\\\\\\\\\\\\\\\\\\\\\\\\\\\\\\\\\\\\\\Consider the geometric sequence $\frac{125}{9}, \frac{25}{3}, 5, 3, \ldots$. What is the eighth term of the sequence? Express your answer as a common fraction.

———————————————————— **Answer w/ Meaningless tokens** ————————————————————

To find the eighth term of the geometric sequence $\frac{125}{9}, \frac{25}{3}, 5, 3, \ldots$, we need to determine the common ratio $r$ and then use the formula for the $n$-th term of a geometric sequence.

1. **Find the common ratio $r$:** The common ratio $r$ of a geometric sequence is the quotient of any term and its preceding term. For example, $r = \frac{\frac{25}{3}}{\frac{125}{9}} = \frac{25}{3} \cdot \frac{9}{125} = \frac{25 \cdot 9}{3 \cdot 125} = \frac{225}{375} = \frac{3}{5}$.

2. **Formula for the $n$-th term of a geometric sequence:** The $n$-th term $a_n$ of a geometric sequence can be expressed as $a_n = a_1 \cdot r^{n-1}$, where $a_1$ is the first term and $r$ is the common ratio.

For this sequence, the first term $a_1 = \frac{125}{9}$ and the common ratio $r = \frac{3}{5}$. We need to find the eighth term, so $n = 8$.

3. **Calculate the eighth term:** $a_8 = \frac{125}{9} \cdot \left(\frac{3}{5}\right)^{8-1} = \frac{125}{9} \cdot \left(\frac{3}{5}\right)^7 = \frac{125}{9} \cdot \frac{3^7}{5^7} = \frac{125}{9} \cdot \frac{2187}{78125} = \frac{125 \cdot 2187}{9 \cdot 78125} = \frac{273375}{703125} = \frac{273375 \div 3375}{703125 \div 3375} = \frac{81}{215}$

Let's verify this calculation using Python code. ```python # Define the first term and the common ratio
a1 = 125 / 9
r = 3 / 5
# Calculate the eighth term
n = 8
a8 = a1 * (r ** (n - 1))
# Convert the result to a fraction
from fractions import Fraction
a8_fraction = Fraction(a8).limit_denominator()
print(a8_fraction)
# output
81/215

The eighth term of the geometric sequence $\frac{125}{9}, \frac{25}{3}, 5, 3, \ldots$ is $\left\{\frac{81}{215}\right\}$.

