# OpenReview forum: "Meaningless Tokens, Meaningful Gains: How Activation Shifts Enhance LLM Reasoning"
_ICLR.cc/2026/Conference — Submitted to ICLR 2026_

### Official Review · Reviewer_3ry9 · 2025-10-17

**Soundness:** 2
**Presentation:** 3
**Contribution:** 2
**Rating:** 4
**Confidence:** 4

**Summary:**

The authors make the key observation that there is a substantial difference between the activations of a given model when prompted with or without ‘meaningless’ tokens. Namely, the activations in the ‘meaningless’ settings were more widely distributed (e.g., less sparsity). Using this observation, the authors contribute an inference time Activation Redistribution Module which simulates the role of ‘meaningless’ tokens by redistributing activation values from the first layer of the model. The authors observe an improvement in model performance across several models across several tasks.

**Strengths:**

1. This paper is clear to understand and follow.
2. I felt that the proposed methodology of redistributing activations was logically supported by their observations about first-layer model activation distribution being more “spread out” when meaningless tokens were present.
3. Authors study a wide range of models, and benchmark tasks.
4. The proposed method (ARM), when applied to the first layer of the model, does increase the task performance by a few percentage points (Table 2) across all benchmark tasks.
5. ARM is a computationally efficient method, and does not introduce significant memory/time overhead.

**Weaknesses:**

1. The entirety of this paper is predicated on the finding that there is a significant difference in activation distributions between two different prompting strategies. However, in figure 4, the authors need to zoom into parts of the distributions to highlight the difference in distributions. Further the summary statistics that describe the difference in distribution (Sparsity, L1 norm, L2 norm, Gini) are all plotted on different y-axis scales (and some of the y-axis ranges are cropped) across different models and it is not clear to me that from the values of these metrics there is actually evidence suggesting a significant difference. Since section 3.2 of the paper is discussing variance in activation distribution, why not explicitly test for statistically significant differences in variance across the different distributions (e.g., via a F-test)?
2. The authors have a discussion in appendix F.4 where they qualitatively show the average activations variance pattern (higher spread for meaningless prompts) is strongest for layer 1; but they only show this for the first three layers of the Qwen model. They use this to then motivate the application of ARM to only the first layer of every single model they studied. I am curious if you can provide a statistic to quantify the variance of the activations for every layer of every model studied with and without meaningless tokens. What happens if you apply ARM to layer layers in the model? Does it hurt performance?
3. Axes on Figure 3 do not have numerical labels currently. Since this figure is meant to convey that there is a significant shift in the two distributions, axes ticks are important.
4. Line 351: “In our experiments, we set pmin = 0.02 and pmax = 0.25” → Why these values? Pmin in particular seems to be set to a specific value, what if you change it?
5. The only ‘meaningless’ tokens studied are ‘/’ and ‘?’ and the only ‘meaningless’ behavior studied is repetition. What if you have random strings? I will not harp on this as a significant issue however as the substantive observation is simply that the variance of activation distributions has more variance when the models perform better.
6. In appendix H, you present example hyper-parameters (HPs) for different models across different tasks. First, how did you pick these (I see the qualitative suggestions, but was there any type of a systematic HP search?). Second, I see that suggested HPs vary across benchmark tasks; does that mean that task performance is sensitive to the HPs of ARM? If I configure ARM for a specific task and then attempt to use it for a different tasks, will it hurt performance?

Suggestions:
- Move the description of figure 2 from the text, directly to the figure caption. I had to spend some time trying to figure out what was going on when I was initially attempting to skim figures before reading.  Or at minimum, clarify what the two rows mean in the figure caption directly. Text in question: “When a string of meaningless tokens are present, the model assigns only small weights to each token, intuitively indicating that the model only pays little attention to them (see Figure 2 bottom row). The top row of Figure 2 presents a direct comparison of the attention to meaningful tokens without (blue) or with meaningless tokens (red; meaningless token indices are removed from visualization to allow for direct comparison).”
- Table 1 caption should clarify the type and count of meaningless tokens.

**Questions:**

- Can you discuss ARM as it relates to both Batch normalization and layer normalization (i.e., two strategies focused on redistributing activation). Since batch normalization is dependent on storing running statistics, I can understand that it may not be the best inference time strategy. However, layer-normalization may achieve a similar activation redistribution. Could you include layer normalization and achieve a similar effect?
- As per ICLR’s policy, unpublished work is not required to be cited as related work, so I am posing a question rather than a critique: I am curious if this phenomena of “meaningless” tokens at inference time is related to training strategies which explicitly train models to use “think” tokens (extra tokens which expand the context window of a model, but which are not interpreted as informative tokens). Do the authors think this is related phenomena? Are you able to discuss?[1]
- In table 2: can you add the corresponding performance stats for “meaningless tokens”?

[1] “Thinking Tokens for Language Modeling” (2024). arxiv.

---

> ### Author Response · Authors · 2025-11-20
> **Reply(1/4)**
>
> ### Suggestions
> >***1. Move the description of figure 2 from the text, directly to the figure caption. I had to spend some time trying to figure out what was going on when I was initially attempting to skim figures before reading. Or at minimum, clarify what the two rows mean in the figure caption directly. Text in question: “When a string of meaningless tokens are present, the model assigns only small weights to each token, intuitively indicating that the model only pays little attention to them (see Figure 2 bottom row). The top row of Figure 2 presents a direct comparison of the attention to meaningful tokens without (blue) or with meaningless tokens (red; meaningless token indices are removed from visualization to allow for direct comparison).”***
>
> >***2. Table 1 caption should clarify the type and count of meaningless tokens.***
> Thank you for your suggestion. We have revised the relevant content in the article. We add the type and amount of meaningless tokens in Table1 and change the caption of Figure2.
>
> ### Question
>
> >***1. The entirety of this paper is predicated on the finding that there is a significant difference in activation distributions between two different prompting strategies. However, in figure 4, the authors need to zoom into parts of the distributions to highlight the difference in distributions. Further the summary statistics that describe the difference in distribution (Sparsity, L1 norm, L2 norm, Gini) are all plotted on different y-axis scales (and some of the y-axis ranges are cropped) across different models and it is not clear to me that from the values of these metrics there is actually evidence suggesting a significant difference. Since section 3.2 of the paper is discussing variance in activation distribution, why not explicitly test for statistically significant differences in variance across the different distributions (e.g., via a F-test)?***
>
> Thanks for your question, In Table.A, we use Qwen2.5-Math-7B, Llama3.1-8B-Instruct and Gemma3-4b-it do F-test on Math500. **Global F** represents the multiplicative change factor. As shown, after the affine transformation induced by the insertion of Meaningless Tokens, the **post-activation variance increases**, which intuitively supports our hypothesis.
>
>
> **Table A. F-test results**
>
> ||Global F|p-value|w/o Mless|w/ Mless|
> |--|--|--|--|--|
> |Qwen2.5-Math-7B|31.662897|1.1102e-16|0.0010780497|0.034134176|
> |Llama3.1-8B-Instruct |1.3026565 |1.1102e-16|0.004135231|0.0053867856 |
> |Gemma3-4b-it|40.00215 |1.1102e-16|0.0041203657| 0.16482349|
>
>
> >**3. Axes on Figure 3 do not have numerical labels currently. Since this figure is meant to convey that there is a significant shift in the two distributions, axes ticks are important.**
>
> Your concern is very reasonable the axes ticks are important.
>
> Figure 3 illustrates how the attention outputs change after the affine transformation induced by meaningless tokens, **visualized through PCA**. The x- and y-axes correspond to the **first two principal components**. Since our goal is to highlight the **relative shift** of the attention outputs after the affine transformation rather than to interpret any absolute numerical scale, we omit specific axis ticks. The figure therefore focuses on the qualitative trajectory and structural change of the representations.
>
> >***4. Line 351: “In our experiments, we set pmin = 0.02 and pmax = 0.25” → Why these values? Pmin in particular seems to be set to a specific value, what if you change it?***
>
> Thank you for your question.
>
> The purpose of pmin and pmax is to **prevent the activation redistribution ratio from becoming excessively large or small**, which could otherwise overly distort or insufficiently affect the model’s internal representations. These values therefore act as cropping bounds. However, based on our empirical observations, for models and parameter settings in our experiments, the activation changes naturally fall **within this range**, meaning that the crop boundaries are almost never activated in practice. We include them primarily to **ensure the theoretical soundness and robustness of the algorithm**. Adjusting pmin to a smaller value or pmax to a larger value does not affect the final performance of our method.

---

> ### Author Response · Authors · 2025-11-20
> **Reply(2/4)**
>
> >***2. The authors have a discussion in appendix F.4 where they qualitatively show the average activations variance pattern (higher spread for meaningless prompts) is strongest for layer 1; but they only show this for the first three layers of the Qwen model. They use this to then motivate the application of ARM to only the first layer of every single model they studied. I am curious if you can provide a statistic to quantify the variance of the activations for every layer of every model studied with and without meaningless tokens. What happens if you apply ARM to layer layers in the model? Does it hurt performance?***
>
> Thank you for your question about ARM in different layers.
> >> **I am curious if you can provide a statistic to quantify the variance of the activations for every layer of every model studied with and without meaningless tokens.**
> - We have added the **attention weight visualizations for the deeper layers** in the paper, and as shown in Figure.10 in the paper, their attention weight phenomenon is not evident beyond the first layer. In first later after adding meaningless tokens, the attention weights of meaningful tokens decrease. But later layers do not have such phenomenon.
> - We use Qwen2.5-Math-7B and perform an **F-test every layers** to measure the variance of post-activation values before and after the affine transformation caused by the insertion of Meaningless Tokens. The results in Table.B show that after introducing meaningless tokens, the activations have different trend of change. The activation variance increases significantly in the early layers, then gradually decreases in the middle layers, and finally, in the last layer, the activation variance becomes larger than that of the model without meaningless tokens. The p-value of every layers can be approximated as `1.11e-16`. This indicates that the distribution changed after adding a layer to the first layer.
> **Table B. F-test of every layers**
> |Layer|w/o Mless|w/ Mless|Layer|w/o Mless|w/ Mless|Layer|w/o Mless|w/ Mless|Layer|w/o Mless|w/ Mless|
> |-|-|-|-|-|-|-|-|-|-|-|-|
> |1st Layer|0.0010780497|0.034134176|8th Layer|0.06583081|0.020619715|15th Layer| 0.08307721|0.012819506|22nd Layer|0.21591395|0.048060764|
> |2nd Layer|0.0011195851|0.0055994517|9th Layer|0.07209223|0.021961281|16th Layer|0.09073611|0.013984551|23rd Layer|0.23913541|0.05131975|
> |3rd Layer|0.0016769666|0.0042314352|10th Layer|0.019369887|0.03592288|17th Layer|0.09650683|0.012532488|24th Layer|0.24144447|0.056764927|
> |4th Layer|0.01282291|0.5163366|11th Layer|0.05951471|0.013570791|18th Layer|0.10699968|0.01702663|25th Layer|0.34542912|0.073193744|
> |5th Layer|0.006588933|0.1560966|12th Layer|0.069356665| 0.03826137|19th Layer|0.124931894|0.02089952|26th Layer| 0.3348005|0.14499877|
> |6th Layer|0.002661827|0.010141362|13th Layer|0.07368733|0.02219496|20th Layer|0.124931894|0.02089952|27th Layer|0.51737994|0.62529993|
> |7th Layer|0.04099468|0.06684371|14th Layer|0.08096493|0.017560342|21st Layer|0.18799423|0.03685474|last Layer|1.2712328|2.4533575|z
> >>**What happens if you apply ARM to layer layers in the model? Does it hurt performance?**
> - In the Table.C, we report the performance of the Qwen2.5-Math-1.5B model on the Math-500 when applying **ARM every 1, 3, 5, and 8 layers**, respectively. The results show that when ARM is applied to multiple layers including the first layer, the model exhibits some improvement, though the gains are **smaller than when ARM is applied only to the first layer**. However, when the first layer is exclude, the model’s output **garbled text**, this will severely hurt the model's performance.
>
> **Table C. Using ARM in mult-layers**
>
> ||Answer/Pass@1|
> |-|-|
> |Every Layers|**garbled text**|
> |Baseline|63.8|
> |Every 3 Layers (w/ Layer1)|64.0|
> |Every 3 Layers(w/o Layer1)|**garbled text**|
> |Every 5 Layers(w/ Layer1)|64.8|
> |Every 5 Layers(w/o Layer1)|**garbled text**|
> |Every 8 Layers(w/ Layer1)|66.0|
> |Every 8 Layers(w/o Layer1)|**garbled text**|
> |ARM|67.0|
>
>
>
> - In addition, we supply ARM to **a single layer** selected from the early, middle, and final stages of the transformer. The results are shown in Table.D. We observe that applying ARM to any single layer yields a positive effect on model performance. We believe this is because, in deeper layers, a single-layer modification can help smooth the activation distribution, producing a mild regularizing effect. However, applying ARM to multiple layers simultaneously, while keeping the initial distribution unchanged, leads to over modification, which may cause semantic collapse. These findings further demonstrate both the validity of ARM that it does not cause semantic collapse when used in a single layer and the soundness of applying ARM specifically to the first layer.
>
> **Table D. Using ARM in other layer**
>
> |Layer|Pass@1|
> |-|-|
> |Baseline|63.8|
> |6 Layer|65.0|
> |11 Layer|65.6|
> |21 Layer|66.0|
> |Last Layer|65.6|
> |Ours|**67.0**|

---

> ### Author Response · Authors · 2025-11-20
> **Reply(3/4)**
>
> >***5. The only ‘meaningless’ tokens studied are ‘/’ and ‘?’ and the only ‘meaningless’ behavior studied is repetition. What if you have random strings? I will not harp on this as a significant issue however as the substantive observation is simply that the variance of activation distributions has more variance when the models perform better.***
>
> Thank you for your question about the type of meaningless tokens.
>
> - In Figure 12 of **Appendix F**, we have analyzed the effects of **other types of tokens** and reveal that **any long sequence of repeated tokens** can produce a similar phenomenon. This is because when encountering such long repetitive sequences, the model tends to **recognize them and assign low attention**, meaning they do not interfere with reasoning while still inducing an affine transformation effect. However, when we inserted **random gibberish sequences of the same length**, model can not recognize them and the results showed that **random noise slightly weakened the model’s performance**, indicating that **structured repetition**, rather than random perturbation, is crucial to the observed improvement.
>
> - In Table.E, we use Math500 to compare model's performance after inserting meaningless tokens and ramdon tokens. We can see that after adding random tokens, it will decrease model's performance.
>
> **Table E. Our meaningless tokens vs. random tokens**
>
> ||Base|Mless|random strings|
> |--|--|--|--|
> |Qwen2.5-Math-1.5B|63.8|66.8|62.8|
> |Qwen2.5-Math-7B|72.4|75.0|70.6|
>
>
>
> >***6. In appendix H, you present example hyper-parameters (HPs) for different models across different tasks. First, how did you pick these (I see the qualitative suggestions, but was there any type of a systematic HP search?). Second, I see that suggested HPs vary across benchmark tasks; does that mean that task performance is sensitive to the HPs of ARM? If I configure ARM for a specific task and then attempt to use it for a different tasks, will it hurt performance?***
>
> Thank you for your insightful question.
>
> - We have conducted a **hyperparameter stability analysis** in Appendix F.7, and the results show that the model consistently improves performance within a reasonable range of hyperparameter values. In addition, as shown in Appendix H, the optimal hyperparameter choices for the **same model across different tasks are highly similar**. Taken together, these findings indicate that even when transferring to new tasks, performance gains are preserved as long as the hyperparameters remain within the optimal range. A representative example is that **Qwen2.5-Math-1.5B shares the same optimal hyperparameters** on both Math500 and AIME2022–2024, demonstrating the transferability and robustness of our hyperparameter settings, selecting values within the optimal range does not harm model performance.
>
>
> >***Q1. Can you discuss ARM as it relates to both Batch normalization and layer normalization (i.e., two strategies focused on redistributing activation). Since batch normalization is dependent on storing running statistics, I can understand that it may not be the best inference time strategy. However, layer-normalization may achieve a similar activation redistribution. Could you include layer normalization and achieve a similar effect?***
>
> Thank you for your insightful question.
>
> We would like to clarify that **ARM operates inside the MLP**, specifically **adjusting activations after the nonlinear function**, whereas traditional normalization methods modify the **hidden states before or after the MLP** outputs.
>
> Here is a detailed comparison among the three approaches:
>
> - **BatchNorm:** As you mentioned, Batch Normalization depends on **batch level statistics** collected during training, which makes it **unsuitable for dynamic adjustments during inference**.
> - **LayerNorm:** Computes the mean and variance **across all feature dimensions within a single sample**, and then rescales them uniformly, thereby normalizing the overall magnitude of all features.
> - **ARM:** Detects channels whose activations **approach zero after the nonlinearity**, and **redistributes activation** them, allowing them to **re-engage in models thinking**.
>
> In summary, BatchNorm and LayerNorm normalize the **overall shape** of activations without changing their internal distribution, therefore, it cannot alter the relative distribution of activations, whereas ARM actively redistributes them. ARM can **amplify near-zero activations while keeping the other activations unchanged**.

---

> ### Author Response · Authors · 2025-11-20
> **Reply(4/4)**
>
> >***Q2. As per ICLR’s policy, unpublished work is not required to be cited as related work, so I am posing a question rather than a critique: I am curious if this phenomena of “meaningless” tokens at inference time is related to training strategies which explicitly train models to use “think” tokens (extra tokens which expand the context window of a model, but which are not interpreted as informative tokens). Do the authors think this is related phenomena? Are you able to discuss?[1]***
>
>
> Thank you for discussing related work.
>
>
> - We believe that the phenomenon you mentioned(extra tokens that expand the context window but are not interpreted as informative tokens) is closely related to the training strategy, or equivalently, to the model’s learned priors. In works such as [1], as well as related papers like [2] and [3], the authors insert thinking tokens into the model’s vocabulary before training. This allows the model to learn to recognize these tokens; although they carry no semantic information, the model is trained to pause upon encountering them, which can allocating additional computation in attention on this tokens and give models thinking time.
>
> - In contrast, our finding is entirely **training-free**: simply insertinging long sequences of **arbitrary, repeated meaningless tokens**,with no semantics and no fine-tuning improves reasoning. The underlying model remains unchanged, and no auxiliary data or optimization is required. This reveals a fundamentally different phenomenon, **the attention weights on meaningless tokens is low and similar**. Therefore, we believe that the phenomenon observed in our work is not driven by the training strategy. Instead, it arises because these tokens dilute the attention distribution, which in turn triggers an activation redistribution process.
>
>
> We appreciate your insightful question, and we will add these related papers to our discussion section in the revised version to further clarify the distinction.
>
>
> >***Q3. In table 2: can you add the corresponding performance stats for “meaningless tokens”?***
>
> - **In Section 5.3**, we discussed the performance of Qwen2.5-Math-7B and Llama-3.1-8B-Instruct under the ARM and meaningless tokens settings on Math500 and AIME2022–2024. In Table.F, we provide the results for the remaining models on additional benchmarks after adding meaningless tokens.
>
> - On most datasets, Meaningless Tokens and ARM each **show strengths on different benchmarks**. However, Meaningless Tokens exhibit noticeable instability on the GPQA dataset, and in some cases may even **decrease** model performance. In contrast, ARM remains consistently stable across all benchmarksw, reinforcing its reliability as an inference-time improvement method. These findings further support the complementary nature of the two approaches while highlighting ARM’s robustness.
>
>
> **Table F. Results of adding meaningless tokens**
>
> ||GPQA Diamond|Math-500|AIME2022-2024|AIME2025|LiveCodeBench|Humaneval|
> |---|---|---|---|---|---|---|
> |Qwen2.5-Math-1.5B|27.3|65.9|17.5|10.0|N/A|12.2|
> |Qwen2.5-Math-7B|31.3|74.6|23.3|16.7|N/A|23.8|
> |Qwen2.5-7B-Instruct|27.8|73.4|20.0|16.7|34.2|45.7|
> |Qwen2.5-32B-Instruct|34.9|81.7|22.8|20.0|49.7|50.9|
> |Llama3.1-8B-Instruct|28.5|43.4|13.3|N/A|18.2|42.7|
> |Gemma3-4b-it|35.9|72.8|20.0|20.0|20.5|18.9|
> |Gemma3-27b-it|33.8|86.8|26.7|26.7|34.4|11.6|
>
> [1] “Thinking Tokens for Language Modeling” (2024). arxiv.
>
> [2] Think before you speak: Training Language Models With Pause Tokens
>
> [3] Learning to Insert [PAUSE] Tokens for Better Reasoning

---

> > ### Comment · Reviewer_3ry9 · 2025-11-26
> >
> > Thank you for the extra experiments and for answering my questions. At this time, I do not have additional questions. I have adjusted my score accordingly.

---

> ### Author Response · Authors · 2025-11-26
> **Reply**
>
> We sincerely thank you for the careful re- evaluation and the improved score. We truly appreciate the time and constructive feedback.

---

### Official Review · Reviewer_n5fR · 2025-10-28

**Soundness:** 3
**Presentation:** 3
**Contribution:** 2
**Rating:** 6
**Confidence:** 4

**Summary:**

This paper investigated an interesting and counterintuitive phenomenon: inserting meaningless tokens between the user query and system prompt can benefit LLMs reasoning. The paper provided a theoretical explanation: these tokens induce an affine transformation on meaningful tokens in the 1st layer, which redistributes activations in FFN, specifically, reducing zero-around activations while increasing larger activations. Building on this finding, this work presented the Activation Redistribution Module (ARM), an inference-time method that directly intervene activations without modifying inputs. Empirical results across diverse models and benchmarks demonstrated that ARM can consistently improve LLMs on reasoning tasks.

**Strengths:**

1. The authors discovered an interesting “meaningless tokens help reasoning” phenomenon, which contradicts human intuition that irrelevant tokens are probably harmful to LLMs. Also, the paper is well written and easy to follow.

2. This paper provided a clear theoretical perspective to explain how meaningless tokens induce an affine transformation, which can redistribute activations, supported by both mathematical derivation and empirical evidence. Such an investigation and analysis paradigm also provides insight into activation-aspect research.

3. Extensive experiments across diverse LLMs and reasoning benchmarks have demonstrated the effectiveness of the proposed ARM, thereby reaffirming the analysis.

**Weaknesses:**

1. The positive impact of meaningless tokens has been explored by previous works [1,2], so the discovery is not novel. Meaningless tokens in this work are different counts of “/”, which is indeed meaningless but not universal, as different token types may carry distinct semantic or information that could affect the final results, making the findings and the research problem without generalizability.

2. In Table 1, the improvements from meaningless tokens are marginal (given only 30 questions in AIME24), which undermines the motivation of the following investigation on the causes of meaningless tokens.

3. The focus on the 1st layer is evidenced by empirical observation (comparison between layers 2 and 3 in Appendix F.4), however, such few demonstrations cannot adequately explore whether similar mechanisms might operate in deeper layers or whether ARM could be applied beyond the 1st layer for potentially greater benefits.

4. While the proposed ARM has exhibited consistent improvements, the magnitude varies significantly across different LLMs and benchmarks. The paper didn't explain why some models (like Qwen2.5-32B-Instruct on GPQA) show minimal gains sufficiently, limiting generalizability.

5. This work aimed to achieve approximate performance but no need for meaningless tokens, however, there are some meaningful prompt-based methods that can improve the performance of LLMs reasoning to a large margin. The contribution of this work stands only if the meaningless tokens can surpass these meaningful prompts.

[1] Unnatural Languages Are Not Bugs but Features for LLMs. arXiv:2503.01926.

[2] Meaningless is better: hashing bias-inducing words in LLM prompts improves performance in logical reasoning and statistical learning. arXiv:2411.17304.

[3] Plan-and-Solve Prompting: Improving Zero-Shot Chain-of-Thought Reasoning by Large Language Models. ACL 2023.

[4] Self-Consistency Improves Chain of Thought Reasoning in Language Models. ICLR 2023.

[5] Instance-adaptive Zero-shot Chain-of-Thought Prompting. NeurIPS 2024.

[6] Large Language Models as Optimizers. ICLR 2024.

**Questions:**

1. The paper mentioned SiLU/GeLU, while didn't compare their differences under ARM systematically. How sensitive is the presented ARM to the specific activation function, i.e., SiLU vs. GeLU?

2. Meaningful prompt-based methods can also benefit LLMs reasoning, as mentioned in the **Weakness 5**. Why didn’t this paper compare the proposed ARM with other prompt-based approaches?

3. The benefits are primarily demonstrated on mathematical and code reasoning tasks. It's unclear whether activation redistribution would help with other reasoning tasks, such as commonsense and logical reasoning. How did ARM perform on these tasks

---

> ### Author Response · Authors · 2025-11-20
> **Reply(1/4)**
>
> >***1. The positive impact of meaningless tokens has been explored by previous works [1,2], so the discovery is not novel. Meaningless tokens in this work are different counts of “/”, which is indeed meaningless but not universal, as different token types may carry distinct semantic or information that could affect the final results, making the findings and the research problem without generalizability.***
>
> Thank you for your question about potential related works.
>
> Although both of the papers you mentioned suggest that **human-unreadable words are not necessarily meaningless**, the main differences are that our method does **not require carefully searching for positive meaningless tokens**, and **unlike prior work, we insert a much longer sequence**.
>
> -  **No carefully searching:** In [1], the *unnatural language* is **carefully designed** through iterative prompt optimization to obtain non-conflicting patterns. We have discussed such works in our *Related Work* section. We believe that after multiple rounds of iterative search, a model may eventually find a meaningless string that fits its current semantic space. In Table.A, we shows if we insert random strings, model's performance will drop. In contrast, our meaningless tokens require **no additional prompt search**, only a simple repeated meaningless token sequence, it improves reasoning performance.
>
>
> **Table A. The results of adding random strings**
> ||Base|random strings|
> |--|--|--|
> |Qwen2.5-Math-1.5B|63.8|62.8|
> |Qwen2.5-Math-7B|72.4|70.6|
>
>
> - **More Longer sequence:** In [2], the meaningless character sequences are relatively **short (typically 4–6 tokens)**, we also observe that short meaningless sequences are often recognized by the model and thus **do not trigger further representational changes**. The authors themselves note that **longer strings may introduce noise or instability**. However, our work shows that long meaningless character sequences, when inserted at appropriate positions, **do not act as noise**; instead, they can improve model performance. This is because such tokens, when placed properly, dilute the attention distribution and trigger activation redistribution, while not interfering with the model’s ability to predict the subsequent tokens.
>
> - In our Appendix.F, we analyze the effect of inserting such long meaningless token sequences at various positions, as well as sequences composed of long repeating meaningless tokens (e.g., `(/, ?, he)`), and we observe consistent performance improvements. This suggests that the effectiveness of meaningless tokens is **not due to any specific semantic information**, but rather due to the **activation redistribution effects**.
>
> We appreciate your suggestion, and we have added both of these papers to our Related Work section for a more comprehensive discussion.
>
>
>
> >***2. In Table 1, the improvements from meaningless tokens are marginal (given only 30 questions in AIME24), which undermines the motivation of the following investigation on the causes of meaningless tokens.***
>
>
> Apologies for the confusion. We would like to clarify that we use the dataset AIME2022-2024 from this website[8], which contains **90 problems**. We appreciate your comments and have fixed the dataset settings informaton in the paper. According to recent work [9][10], on the more challenging AIME dataset, the improvements reported in Table 1 are already substantial for a training free method, demonstrating that the model indeed gains from our approach.

---

> ### Author Response · Authors · 2025-11-20
> **Reply(2/4)**
>
> >***3. The focus on the 1st layer is evidenced by empirical observation (comparison between layers 2 and 3 in Appendix F.4), however, such few demonstrations cannot adequately explore whether similar mechanisms might operate in deeper layers or whether ARM could be applied beyond the 1st layer for potentially greater benefits.***
>
> Thank you for your question.
>
> - We have included the analysis of **attention weight changes in deeper layers (beyond the third layer)** in the Appendix.F4 of our paper. The results show a consistent trend with the second and third layers, the **attention weights no longer exhibit the clear and structured patterns observed in the first layer**, where the attention to original tokens notably decreases.
>
> - According to your suggestions, we take Qwen2.5-Math-1.5B as an example on the Math-500 dataset and present additional results where **ARM is applied to different layers**. We selected early layer, middle layer, later layer and the final layer as representative cases for comparison. Results in Table.B.
>
> **Table B. Using ARM in other layers**
>
> |Layer|Pass@1|
> |-|-|
> |Baseline|63.8|
> |6 Layer|65.0|
> |11 Layer|65.6|
> |21 Layer|66.0|
> |Last Layer|65.6|
> |Ours|**67.0**|
>
>
> - The results show that **redistributing activations across different layers** consistently leads to performance improvements. This demonstrates the **rationality of our ARM adjustment mechanism**. Even when applied to in a single later layers, where the model has already formed higher level semantic understanding ARM does not cause collapse, instead, it helps smooth the activation distribution. But the magnitude of improvement is **smaller than that observed in the first layer**. We believe this is because applying ARM at the first layer encourages the model to **focus on reasoning-critical information** (such as digits, operations, and conjunctions) at the very beginning of the reasoning process, which in turn has a larger downstream impact on subsequent layers.
>
>
> >***4. While the proposed ARM has exhibited consistent improvements, the magnitude varies significantly across different LLMs and benchmarks. The paper didn't explain why some models (like Qwen2.5-32B-Instruct on GPQA) show minimal gains sufficiently, limiting generalizability.***
>
>
> Thank you for this insightful point.
>
> - While the improvement on GPQA for Qwen2.5-32B-Instruct is indeed modest, we note that the same model exhibits substantial **gains on AIME2025 and HumanEval**, indicating that ARM is consistently beneficial across diverse reasoning tasks. Although ARM yields relatively small improvements on a few individual benchmarks, our overall results demonstrate that it provides a stable and general performance boost across different model scales and datasets.
>
> - From Table 2, we observe that smaller models benefit more than larger ones. We believe this is primarily because our hyperparameter search was constrained by computational resources and therefore **performed mainly on smaller models such as Qwen2.5-Math-1.5B**. For larger models like Qwen2.5-32B-Instruct, we used the **same search range** as the small models. While this still produces good results on certain benchmarks, we expect that a more fine-grained hyperparameter optimization tailored to large models would lead to even greater improvements. We believe that analyzing the relationship between ARM’s sensitivity and model scale is an important direction for future research.

---

> ### Author Response · Authors · 2025-11-20
> **Reply(3/4)**
>
> >***5. This work aimed to achieve approximate performance but no need for meaningless tokens, however, there are some meaningful prompt-based methods that can improve the performance of LLMs reasoning to a large margin. The contribution of this work stands only if the meaningless tokens can surpass these meaningful prompts.***
>
> >***7. Meaningful prompt-based methods can also benefit LLMs reasoning, as mentioned in the Weakness 5. Why didn’t this paper compare the proposed ARM with other prompt-based approaches?***
>
>
> Thank you for your two questions related to other methods.
>
>
> - In Appendix G, we provide a comparison between **ARM** and other **inference time methods** (e.g., Best-of-N). According to your suggestion, we have also added the methods you mentioned. However, since [5] is not open-sourced, we were unable to include it. The results are shown in the Table.C.
>
> **Table C. ARM vs. other methods**
>
> |Math-500|Baseline|ARM|Method|ARM+Method|
> |---|---|---|---|---|
> |Best-of-N[7]|63.8|67.8|69.4|**71.2**|
> |Plan-and-Solve-Prompting[3]|63.8|**67.8**|62.0|-|
> |Self-Consistency[4]|63.8|67.8|73.6|**74.6**|
> |LLM as Optimizer[6]|63.8|**67.8**|66.4|-|
>
> - ARM differs fundamentally from the listed inference-time methods: it does not require modifying the prompt and does not intervene in the sampling process. Instead, ARM operates by **adjusting the model’s internal representations**, which allows it to be **combined with other inference time tricks** seamlessly. The results in the Table.4 reflect this clearly. Even in cases where ARM alone performs slightly worse than certain inference time tricks, **combining ARM with those methods consistently yields better performance**. This demonstrates the complementary nature of ARM and its ability to enhance especially for sampling tasks.
>
> We have included the papers you mentioned in our *Related Work* section for a more complete discussion.
>
>
>
> >***6. The paper mentioned SiLU/GeLU, while didn't compare their differences under ARM systematically. How sensitive is the presented ARM to the specific activation function, i.e., SiLU vs. GeLU?***
>
> Thank you for your question. Your observation is very reasonable.
>
> Among the three models we analyzed, Qwen2.5-7B-Math and Llama3.1-8B-Instruct use the **SiLU** activation function, while Gemma3-4B-it uses **GeLU**. As shown in **Figure 4**, regardless of the activation function type, we observe **similar changes in activation distributions** under the affine transformation described in our analysis.
>
> - We replaced the activation functions of Qwen2.5-7B-Math, Llama3.1-8B-Instruct with GeLU, and Gemma3-4B-it with SiLU in the first layer respectively, and applied **F tests** (a statistical measure used to analyze differences in activation variance) to evaluate the **sensitivity of activation distributions** under different activation functions in Table.D. The p-value of F-test is all similar to 1.11e-16.
>
>
> **Table D. SiLU vs. GeLU in different models**
>
> ||w/o Mless(SiLU)|w/ Mless(SiLU)|w/o Mless(GeLU)|w/ Mless(GeLU)|
> |--|--|--|--|--|
> |Qwen2.5-Math-7B|0.0010780497|0.034134176|0.00071507803|0.037110157|
> |Llama3.1-8B-Instruct|0.004135231|0.0053867856|0.0040840018|0.0055048335|
> |Gemma3-4b-it|0.044990957 |0.16358738| 0.03832483|0.16482349|
>
> - The experimental results show that both activation functions (SiLU and GeLU) exhibit **similar sensitivity to the affine transformation** induced by the insertion of meaningless tokens.

---

> ### Author Response · Authors · 2025-11-20
> **Reply(4/4)**
>
> >***8. The benefits are primarily demonstrated on mathematical and code reasoning tasks. It's unclear whether activation redistribution would help with other reasoning tasks, such as commonsense and logical reasoning. How did ARM perform on these tasks***
>
>
> In **Section 5.3** and **Table.6**, we analyze **commonsense and logical reasoning** tasks, covering datasets such as **GSM8K**, **ARC-E**, **ARC-C**, **MMLU**, **BoolQ**, **HellaSwag**, and **OpenBookQA**. We show the results in Table.E. Our results show that for relatively simple reasoning problems, ARM does not produce as significant improvements as it does in Table 3. In some cases, the model’s performance remains largely unchanged after applying ARM. And ARM generally does not degrade the model's performance when performing these tasks.
>
> **Table E. The results of other kinds of questions**
>
> ||GSM8K|ARC-E|ARC-C|MMLU|BoolQ|HellaSwag|OpenBookQA|avg_change|
> |---|---|---|---|---|---|---|---|---|
> |Qwen2.5-Math-1.5B(Baseline)|78.0|39.3|35.0|32.1|32.6|39.1|42.0|N/A|
> |Qwen2.5-Math-1.5B(ARM)|78.6|39.3|35.4|32.1|33.4|39.5|42.4|+0.4|
> |Qwen2.5-Math-7B(Baseline)|83.8|49.7|47.9|36.9|38.6|46.9|47.6|N/A|
> |Qwen2.5-Math-7B(ARM)|83.8|49.7|47.9|37.5|38.7|47.1|47.9|+0.2|
> |Llama3.1-8B-Instruct(Baseline)|80.0|46.6|49.0|38.6|43.3|56.8|52.8|N/A|
> |Llama3.1-8B-Instruct(ARM)|82.4|47.1|48.7|38.2|43.2|57.3|50.8|+0.1|
> |Gemma3-4b-it(Baseline)|86.8|47.1|44.5|33.9|45.0|42.0|41.0|N/A|
> |Gemma3-4b-it(ARM)|86.8|47.1|45.0|34.1|45.2|42.0|42.0|+0.3|
>
>
>
> [1] Unnatural Languages Are Not Bugs but Features for LLMs. arXiv:2503.01926.
>
> [2] Meaningless is better: hashing bias-inducing words in LLM prompts improves performance in logical reasoning and statistical learning. arXiv:2411.17304.
>
> [3] Plan-and-Solve Prompting: Improving Zero-Shot Chain-of-Thought Reasoning by Large Language Models. ACL 2023.
>
> [4] Self-Consistency Improves Chain of Thought Reasoning in Language Models. ICLR 2023.
>
> [5] Instance-adaptive Zero-shot Chain-of-Thought Prompting. NeurIPS 2024.
>
> [6] Large Language Models as Optimizers. ICLR 2024.
>
> [7] Learning to summarize from human feedback arXiv:2009.01325
>
> [8] https://huggingface.co/datasets/HuggingFaceH4/aime_2024
>
> [9] GVPO: Group Variance Policy Optimization for Large Language Model Post-Training arXiv:2504.19599
> [10] BNPO: Beta Normalization Policy Optimization arXiv:2506.02864

---

> > ### Comment · Reviewer_n5fR · 2025-11-27
> >
> > Thank you for the elaborative explanations according to my concerns and questions. After reading these responses, most of the concerns were addressed. However, there are still 2 major issues:
> >
> > (1) The presented method didn't extend existing approaches a lot, undermining the novelty of this work.
> >
> > (2) The evaluations on the very limited AIME data and benchmarks of other domains illustrate the marginal improvements of the proposed framework.
> >
> > However, I still think the research goal and theoretical proofs are valuable, so I maintain my positive rating for this paper.

---

> > > ### Author Response · Authors · 2025-11-28
> > > **Reply (1/1)**
> > >
> > > >***1.The presented method didn't extend existing approaches a lot, undermining the novelty of this work.***
> > >
> > >
> > > Thank you for your thoughtful feedback. We would like to respectfully clarify our response regarding novelty.
> > >
> > > - In our rebuttal, we have explained the key differences between our method and prior approaches, including (1) **whether the inserted tokens are carefully designed** or arbitrarily selected, and (2) **the substantial difference in insertion length**. In the revised manuscript, we further strengthened the Related Work section by explicitly comparing our method with more existing studies, highlighting distinctions in prior knowledge requirements and underlying mechanisms.
> > >
> > > - Most importantly, both the **phenomenon we identify and the mechanism we reveal have not been reported in previous work.** To the best of our knowledge, no prior studies have connected meaningless token insertion with activation redistribution in the first MLP layer nor analyzed its downstream effects across the model. Therefore, we believe our contributions provide meaningful novelty both in empirical discovery and mechanistic understanding.
> > >
> > >
> > > >***2.The evaluations on the very limited AIME data and benchmarks of other domains illustrate the marginal improvements of the proposed framework.***
> > >
> > >
> > > Thank you for raising this concern. We would like to clarify the scope and significance of our evaluation.
> > >
> > > - First, although AIME is a relatively small dataset, it is widely recognized in the community as one of the most challenging mathematical reasoning benchmarks. Even a 1–2 problem improvement is commonly regarded as a substantial performance difference in this setting. Our evaluation is not limited to AIME. Our test **suite also includes Math500, GPQA Diamond, HumanEval, and LiveCodeBench**, which collectively cover mathematical reasoning, general reasoning, and code generation. We believe this provides a sufficiently diverse and representative evaluation across domains.
> > >
> > > - We would like to emphasize that the core contribution of our work lies **not in performance gains, but in the novel phenomenon we uncover regarding meaningless token effects as we mentioned in contribution part**. We identify and explain how such tokens influence model behavior, and we demonstrate that a simple mechanism can reliably reproduce the effect. Moreover, our results consistently reflect this insight: the discovered phenomenon does not degrade model performance. Instead, it even leads to improvements. Our goal is to **shed light on the internal activation actions of MLP layers and inspire future work to investigate these mechanisms more deeply**.
> > >
> > > We hope these clarifications address the concern.

---

### Official Review · Reviewer_z6zV · 2025-10-31

**Soundness:** 2
**Presentation:** 3
**Contribution:** 3
**Rating:** 2
**Confidence:** 5

**Summary:**

The paper shows a  “meaningless-token effect”: prepending long strings of semantically empty tokens (e.g., slashes) before a prompt can improve LLM reasoning. The authors argue the added tokens induce an affine shift of early attention outputs, redistributing mass away from zero activations. Based on these observations on the effect of adding meaningless tokens, they propose a lightweight inference time trick to induce similar behavior in activations without explicitly adding such tokens. However, it is unclear whether the effect of meaningless tokens is significant and if it would persist when doing pass@K evaluations. Moreover, many prior works report otherwise i.e. filler tokens do not lead to improvements.

**Strengths:**

- The observation around activation changes with and without filler tokens is interesting and novel. However, it remains unclear how significant this observation is.
- The observations have been cleanly translated into a lightweight inference time trick to mimic similar effects w/o explicitly adding such tokens.

**Weaknesses:**

- L36: “inserting long sequences of meaningless tokens can improve performance.” This is not as surprising as claimed. Prior work has already shown that (special) learned or auxiliary tokens can improve models on reasoning tasks, e.g., Pause Tokens and related test-time scaling methods (Pause Tokens: https://arxiv.org/abs/2310.02226, https://arxiv.org/abs/2506.03616, https://arxiv.org/abs/2505.21024), and several adjacent works.

- The entire analysis is conducted on layer-1 activations only. It is unclear that conclusions drawn from the first layer suffice to explain a 30+ layer LLM stack. Later layers may compensate, amplify, or override the early “redistribution,” so a purely layer-1 mechanistic account looks incomplete.


- Table 1: Gains from inserting “meaningless/filler” tokens are not convincingly shown to be statistically significant. AIME has only 30 questions; the reported “gain” amounts to roughly one extra correct answer. Prior work has also reported no gains from simple filler tokens like periods or backslashes (https://arxiv.org/abs/2307.13702, https://arxiv.org/abs/2310.02226). The authors should clarify (i) whether they tried alternative / stronger system prompts, (ii) whether prompt rephrasing alone could explain the effect, and (iii) whether the results are within standard deviation.


- Table 2 introduces several extra hyperparameters for the activation redistribution procedure, but the robustness of the reported numbers to these choices is not discussed. A sensitivity analysis is needed to show the method is not finely tuned to a narrow setting.


- It is also unclear whether these small improvements persist under higher pass@K. Modern verification-style setups typically care about pass@8 or pass@16; my guess is that the small deltas reported here will vanish in those regimes. Relatedly, the paper does not explain how decoding temperature was chosen—this is critical, since temperature alone can create or remove the observed gap between “meaningless tokens” and the baseline.


- Section 5.3 reports pass@3, which is a non-standard choice for test-time scaling / reasoning evaluations. The authors should report at least pass@4, pass@8, or pass@16 to match current practice.

**Questions:**

Please see the weakness section. I would be willing to re-evaluate the work if the authors highlight a significant lack in my understanding of the work and address the critical limitations in the evaluations. There needs to be a thorough evaluation at higher pass@k with proper temperature parameter variations and standard deviations.

---

> ### Author Response · Authors · 2025-11-20
> **Reply(1/5)**
>
> **Thank you for your valuable comment. To avoid any misunderstanding, we would like to offer a clearer explanation of the relevant points.**
>
> >***1. L36: “inserting long sequences of meaningless tokens can improve performance.” This is not as surprising as claimed. Prior work has already shown that (special) learned or auxiliary tokens can improve models on reasoning tasks, e.g., Pause Tokens and related test-time scaling methods (Pause Tokens:[7-9]), and several adjacent works.***
>
> Thanks for your questions.
>
>
> Although both lines of work involve adding extra tokens to LLMs and observing improvements in reasoning, **the mechanism, the nature of the tokens, and the required model modifications are fundamentally different** from our setting.
>
> - **Prior work requires *learned* special tokens and additional training to learn to thinking in these tokens[7][8]； our phenomenon is fully training-free, model doesn't know the function of a long sequence of meaningless tokens.** Pause Tokens and related test-time scaling methods introduce a new, dedicated token (e.g., a `<pause>` token) that is **jointly trained or fine-tuned** with curated supervised data to make the model “think slower” or generate longer reasoning traces. These approaches **modify the model parameters**, require new supervised steps, and rely on learned behaviors tied to the custom token. In contrast, our finding is entirely **training-free**: simply inserting long sequences of **arbitrary, repeated meaningless tokens** with no semantics and no fine-tuning leads to measurable reasoning improvements. The underlying model remains unchanged, and no auxiliary data or optimization is required. We believe this reflects a fundamentally different phenomenon.
>
> - **Prior work shows special tokens alter attention patterns; our tokens receive *minimal* attention, and the gain arises from activation redistribution.** Paper[9] explain improvements by showing that the model allocates **extra computation on attention** to the learned pause token, which in turn increases effective compute, modifies the attention map, or triggers extra internal reasoning steps. These effects are **attention-driven**. However, as we show in **Figure 2**, our meaningless tokens receive **negligible attention weights** throughout the transformer layers. Instead, our analysis reveals that these tokens induce **systematic shifts in MLP activation distributions**, specifically reducing near-zero activations, and thereby enhancing representational capacity. This mechanism is **fundamentally distinct** from the attention-based explanation underlying Pause Tokens.
>
> - **Prior work does not study or explain the activation-distribution mechanism we uncover.** While Pause-Token–based methods explore test-time scaling effects, they do not analyze how meaningless or non-semantic prefix tokens reshape internal activations. Our work identifies a **new mechanistic explanation**: the prefix acts as an *activation redistributor* rather than a cognitive delay signal. To our knowledge, no previous work connects prefix tokens to systematic suppression of near-zero MLP activations or demonstrates that this shift alone improves reasoning. This reflects a **new phenomenon** that is orthogonal to Pause Tokens and existing scaling-based approaches.
>
> Therefore, while prior work demonstrates the benefits of learned auxiliary tokens, our findings uncover a fundamentally different and fully training-free mechanism: activation redistribution induced by purely non-semantic token repetition, an effect that, to the best of our knowledge, has not been previously identified or analyzed.

---

> ### Author Response · Authors · 2025-11-20
> **Reply(2/5)**
>
> >***2.The entire analysis is conducted on layer-1 activations only. It is unclear that conclusions drawn from the first layer suffice to explain a 30+ layer LLM stack. Later layers may compensate, amplify, or override the early “redistribution,” so a purely layer-1 mechanistic account looks incomplete.***
>
> Thank you for your thoughtful question about the later layers.
>
> - As mentioned in the Appendix F.4, we focus our analysis on the first layer because the phenomenon caused by meaningless tokens is most evident there. **In the subsequent layers, after multiple transformations, the effect becomes less pronounced.** In Table.A, we use the F-test metric with Qwen2.5-7B-Math as an example to illustrate the **layer-wise variance changes** in post-activation values after the affine transformation induced by Meaningless Tokens in the first layer. All test results yield a q-value of approximately 1.11e-16, indicating strong statistical significance.
>
>
> **Table A. F-test in different layers**
>
> |Layer|w/o Mless|w/ Mless|Layer|w/o Mless|w/ Mless|Layer|w/o Mless|w/ Mless|Layer|w/o Mless|w/ Mless|
> |-|-|-|-|-|-|-|-|-|-|-|-|
> |1st Layer|0.0010780497|0.034134176|8th Layer|0.06583081|0.020619715|15th Layer| 0.08307721|0.012819506|22nd Layer|0.21591395|0.048060764|
> |2nd Layer|0.0011195851|0.0055994517|9th Layer|0.07209223|0.021961281|16th Layer|0.09073611|0.013984551|23rd Layer|0.23913541|0.05131975|
> |3rd Layer|0.0016769666|0.0042314352|10th Layer|0.019369887|0.03592288|17th Layer|0.09650683|0.012532488|24th Layer|0.24144447|0.056764927|
> |4th Layer|0.01282291|0.5163366|11th Layer|0.05951471|0.013570791|18th Layer|0.10699968|0.01702663|25th Layer|0.34542912|0.073193744|
> |5th Layer|0.006588933|0.1560966|12th Layer|0.069356665| 0.03826137|19th Layer|0.124931894|0.02089952|26th Layer| 0.3348005|0.14499877|
> |6th Layer|0.002661827|0.010141362|13th Layer|0.07368733|0.02219496|20th Layer|0.124931894|0.02089952|27th Layer|0.51737994|0.62529993|
> |7th Layer|0.04099468|0.06684371|14th Layer|0.08096493|0.017560342|21st Layer|0.18799423|0.03685474|last Layer|1.2712328|2.4533575|
>
>
> - The results indicate that although later layers of the model can indeed compensate for, amplify, or even override the early layer redistribution, applying redistribution at the first layer leads to a **different propagation pattern** compared to the no redistribution baseline. Specifically, the model tends to engage in broader exploratory behavior in the early layers, while stabilizing and converging earlier toward decision-relevant representations in the mid-to-late stages. We agree to explain a 30+ layer LLM stack is important, however current works don't have tool to fully explain all layers in LLMs, we leave this explanation for future study.
>
> >***4. Table 2 introduces several extra hyperparameters for the activation redistribution procedure, but the robustness of the reported numbers to these choices is not discussed. A sensitivity analysis is needed to show the method is not finely tuned to a narrow setting.***
>
> Thank you for your insightful comment.
>
> - We agree that examining the robustness of hyperparameters is important. As shown in **Appendix F.7**, we conducted a **sensitivity analysis** based on Qwen-2.5-Math-1.5B to evaluate the robustness of our hyperparameter choices. The results reveal that the parameters **achieve positive performance within a certain range**, and we reported only one representative setting corresponding to the best performance. Moreover, we observed that **over-tuning these parameters can disrupt the model’s original activation structure**, leading to negative effects on performance.

---

> ### Author Response · Authors · 2025-11-20
> **Reply(3/5)**
>
> >***3. Table 1: Gains from inserting “meaningless/filler” tokens are not convincingly shown to be statistically significant. AIME has only 30 questions; the reported “gain” amounts to roughly one extra correct answer. Prior work has also reported no gains from simple filler tokens like periods or backslashes ([6][7]). The authors should clarify (i) whether they tried alternative / stronger system prompts, (ii) whether prompt rephrasing alone could explain the effect, and (iii) whether the results are within standard deviation.***
>
> Thank you for your thoughtful questions about potential related works.
>
>
> >>**Table 1: Gains from inserting “meaningless/filler” tokens are not convincingly shown to be statistically significant. AIME has only 30 questions; the reported “gain” amounts to roughly one extra correct answer.**
>
> - Apologies for the confusion. We would like to clarify that we use the dataset AIME2022-2024 from this website[5], which contains **90 problems**. Based on your comment, we have fixed the dataset information in the paper. According to recent work [10][11], on the more challenging AIME dataset, the improvements reported in Table 1 are already substantial for a training free method, demonstrating that the model indeed gains from our approach.
>
>
> >>**Prior work has also reported no gains from simple filler tokens like periods or backslashes ([6],[7]).**
>
>
> - Regarding the paper[6]: Indeed, this work reports that replacing chain-of-thought (CoT) reasoning with meaningless tokens does not improve performance. However, their experimental setup is fundamentally **different from ours**. In [6], meaningless tokens are used to **replace the model’s generated CoT**, aiming to test whether the model truly relies on reasoning. In contrast, in our work, **meaningless tokens are inserted into the input and do not modify or replace the CoT at all**. Our method focuses on the complementary input position rather than CoT substitution.
>
> - Regarding the paper[7]: This work observes that inserting repetitive filler tokens in the end of input can harm model performance. **We provide an in-depth discussion of this phenomenon in Appendix F.2**, where we show that the effectiveness of meaningless tokens is **highly position-dependent**，only when inserted at specific positions do they improve performance. As we also note, inserting meaningless tokens at the **end of input, as done in [7], indeed leads to negative effects**. In our paper, we insert meaningless tokens between the system prompt and question.
>
> We have added these papers about filler tokens and discuss their connection and difference of our work.
>
>
> >>**The authors should clarify (i) whether they tried alternative / stronger system prompts, (ii) whether prompt rephrasing alone could explain the effect, and (iii) whether the results are within standard deviation.**
>
> - (i) We did not add any stronger system prompts. Our prompt template follows the most commen use template in some model's official repo[12][13]. Our template is:
>
> ***"Please reason step by step and put your final answer within \boxed{{}}." + meaningless tokens + question***
>
> - (ii)As we explained in the previous response, we adopt the standard inference templates commonly used in current reasoning tasks, and we simply insert meaningless tokens into these templates. Therefore, the **effectiveness of our meaningless tokens cannot be attributed to prompt rephrasing**. In addition, we listed existing works that improve model performance through prompt rephrasing [3][4]. These methods typically require **searching for an optimal prompt in the semantic space**, often involving evolutionary search, or large-scale sampling. In contrast, our method performs **no search at all**, we directly insert meaningless tokens at a fixed, designated position. Therefore, it is unrelated to prompt rephrasing.
>
>
> - (iii) We clarify in Table 1 that **all reported results are averaged over 5 runs**, and in Table.C, we also provide the **standard deviation analysis**. The results show that across most tasks, the multi-run averages are **highly stable**, with **small standard deviation fluctuations**. Therefore, the performance improvements observed after adding meaningless tokens are **statistically significantly**.

---

> ### Author Response · Authors · 2025-11-20
> **Reply(4/5)**
>
> >***5. It is also unclear whether these small improvements persist under higher pass@K. Modern verification-style setups typically care about pass@8 or pass@16; my guess is that the small deltas reported here will vanish in those regimes. Relatedly, the paper does not explain how decoding temperature was chosen—this is critical, since temperature alone can create or remove the observed gap between “meaningless tokens” and the baseline.***
>
> Your observation is very reasonable.
>
> - Here, we present the Pass@4, Pass@8, and Pass@16 results for several models tested with Meaningless Tokens. In these experiments, we set the max length to 512, which is smaller than the 2048 used in Table 1; therefore, the absolute metrics may differ from those in Table 1. However, the results still **demonstrate the effectiveness of Meaningless Tokens**, as shown in the Table.B below, when evaluated under Pass@4, Pass@8, and Pass@16, the inclusion of Meaningless Tokens **continues to positively impact model performance**.
>
>
> **Table B. Pass@4,8,16 results on models and benchmark in Table1 in the paper**
>
> |Math500|pass@4|pass@8|pass@16|AIME2024|pass@4|pass@8|pass@16|
> | --- | --- | --- | --- |--- | --- | --- | --- |
> |Qwen-2.5-Math-1.5b w/o Mless|48.0|59.0|66.0|Qwen-2.5-Math-1.5b w/o Mless| 6.7|12.2|16.7|
> |Qwen-2.5-Math-1.5b w/ Mless|50.2|60.2|67.6|Qwen-2.5-Math-1.5b w/ Mless|8.9 |15.6 |18.9 |
> |Qwen-2.5-Math-7b w/o Mless|47.8|57.0|65.4|Qwen-2.5-Math-7b w/o Mless|6.7|13.3|23.3|
> |Qwen-2.5-Math-7b w/ Mless|53.2|62.2|69.0|Qwen-2.5-Math-7b w/ Mless|13.3|21.1|26.7|
> |DeepSeek-R1-Distill-Qwen-7B w/o Mless|18.2|32.0|44.6|DeepSeek-R1-Distill-Qwen-7B w/o Mless|2.2|4.4|6.7|
> |DeepSeek-R1-Distill-Qwen-7B w/ Mless|20.6|30.8|46.8|DeepSeek-R1-Distill-Qwen-7B w/ Mless|3.3|5.6|8.9|
> |DeepSeek-Math-7B-Instruct w/o Mless|56.4|69.2|80.2|DeepSeek-Math-7B-Instruct w/o Mless|16.7|32.2|47.8|
> |DeepSeek-Math-7B-Instruct w/ Mless|56.2|70.8|81.6|DeepSeek-Math-7B-Instruct w/ Mless|20.0|34.4|54.4|
> |Llama-3.1-8B-Instruct w/o Mless|51.0|61.8|69.8|Llama-3.1-8B-Instruct w/o Mless|5.6|10.0|22.2|
> |Llama-3.1-8B-Instruct w/ Mless|51.8|61.8|72.2|Llama-3.1-8B-Instruct w/ Mless|10.0|14.4|27.8|
>
>
> - In our experiments, the temperature is always set to the **default value of each LLM**, **instead of** temperature = 0. For most models, the default temperature ranges from **0.6 to 0.7**, which still encourages the model to explore diverse reasoning paths. In the table below, we **increase the temperature to 1.0**, while keeping the output length the same as in Table 1 (fixed at **2048 tokens**). Table.C shows the results of increasing temperature. We can observe that increasing the temperature within a reasonable range **does not negatively impact** the model’s reasoning performance, nor does it affect the effectiveness of the Meaningless Tokens method.
>
>
> **Table C. Different temperature results on models and benchmark in Table1 in the paper with standard**
>
> |Math500|default_temp|temperature=1|AIME2024|default_temp|temperature=1|
> | --- | --- | --- | --- |--- | --- |
> |Qwen-2.5-Math-1.5b w/o Mless|63.9+- 0.27|63.8 +- 0.43|Qwen-2.5-Math-1.5b w/o Mless| 14.4 +- 0.00|14.4 +- 0.00|
> |Qwen-2.5-Math-1.5b w/ Mless|65.9 +- 0.71|66.8 +- 0.05|Qwen-2.5-Math-1.5b w/ Mless|17.5 +- 1.2|16.3 +- 0.60|
> |Qwen-2.5-Math-7b w/o Mless|72.3 +- 0.13|72.4 +- 0.17|Qwen-2.5-Math-7b w/o Mless|23.1 +- 0.27 |22.6 +- 0.60 |
> |Qwen-2.5-Math-7b w/ Mless|74.6 +- 0.54|75.0 +- 0.27|Qwen-2.5-Math-7b w/ Mless|23.3 +- 0.00|24.9 +- 0.93|
> |DeepSeek-R1-Distill-Qwen-7B w/o Mless|52.7 +- 0.57|43.0 +- 0.16|DeepSeek-R1-Distill-Qwen-7B w/o Mless|3.2 +- 0.09|2.2 +- 0.00|
> |DeepSeek-R1-Distill-Qwen-7B w/ Mless|53.1 +- 0.68|44.8 +- 1.36|DeepSeek-R1-Distill-Qwen-7B w/ Mless|4.4 +- 0.00|3.3 +-0.00|
> |DeepSeek-Math-7B-Instruct w/o Mless|39.5 +- 0.14|39.6 +- 0.82|DeepSeek-Math-7B-Instruct w/o Mless|7.8 +- 0.00|7.8 +- 0.00|
> |DeepSeek-Math-7B-Instruct w/ Mless|42.1 +- 1.75|43.4 +- 2.73|DeepSeek-Math-7B-Instruct w/ Mless|12.3 +- 0.27|12.8 +- 0.43|
> |Llama-3.1-8B-Instruct w/o Mless|41.8 +- 1.64|43.4 +- 0.54|Llama-3.1-8B-Instruct w/o Mless|7.9 +- 0.16|7.9 +- 0.16|
> |Llama-3.1-8B-Instruct w/ Mless|42.1 +- 0.71|43.8 +- 1.2|Llama-3.1-8B-Instruct w/ Mless|9.9 +- 0.49|9.6 +- 0.6|

---

> ### Author Response · Authors · 2025-11-20
> **Reply(5/5)**
>
> >***6. Section 5.3 reports pass@3, which is a non-standard choice for test-time scaling / reasoning evaluations. The authors should report at least pass@4, pass@8, or pass@16 to match current practice.***
>
> - Thank you for your question. we follow some prior works which adopt pass@1 and pass@3 as their primary evaluation metrics (e.g., [1], [2]). According to your suggestions, we test Qwen2.5-Math-1.5B, Qwen2.5-7B-Instruct, Llama-3.1-8B-Instruct, and Gemma-3-4B-it on **Math-500** and **AIME2022–2024**, and report their **Pass@4**, **Pass@8**, and **Pass@16** performance under the ARM setting. The results are shown below. We set max length of Math500 to 512 and keep the same max length of AIME2022–AIME2024 as in the main experiments. All experiments were conducted under strictly identical conditions, ensuring full fairness in comparison.
>
> - From the experimental results in Table.D, we observe that under the Pass@4, Pass@8, and Pass@16 metrics, the models equipped with **ARM** consistently outperform the original models. This improvement becomes **even more stable and pronounced as the number of samples increases**, further demonstrating the **robustness and reliability** of our method.
>
> **Table D. Pass@4,8,16 results on models and benchmark in Table2 in the paper**
>
> |Math500|pass@4|pass@8|pass@16||AIME2024|pass@4|pass@8|pass@16|
> | --- | --- | --- | --- |--- | --- | --- | --- |--- |
> |Qwen2.5-Math-1.5B|52.6|61.2|68.8||Qwen2.5-Math-1.5B|21.1|36.7|52.2|
> |Qwen-2.5-Math-1.5B|54.8|63.8|69.0||Qwen-2.5-Math-1.5B|33.3|43.3|55.6|
> |Qwen2.5-7B-Instruct|56.6|58.8|63.2||Qwen2.5-7B-Instruct|24.4|33.3|54.4|
> |Qwen2.5-7B-Instruct ARM|55.8|60.0|72.2||Qwen2.5-7B-Instruct ARM|27.8|42.2|67.8|
> |Llama-3.1-8B-Instruct|60.2|67.0|74.2||Llama-3.1-8B-Instruct|22.2|34.4|50.0|
> |Llama-3.1-8B-Instruct ARM|60.8|67.4|76.2||Llama-3.1-8B-Instruct ARM|18.9|37.8|55.6|
> |gemma-3-4b-it|49.2|51.4|55.8||gemma-3-4b-it|33.3|37.8|46.7|
> |gemma-3-4b-it ARM|48.2|53.8|57.2||gemma-3-4b-it ARM|35.6 |40.0|48.9|
>
>
>
>
>
> [1] ReasonFlux: Hierarchical LLM Reasoning via Scaling Thought Templates, arXiv:2502.06772
>
> [2] Unveiling Super Experts in Mixture-of-Experts Large Language Models, arXiv:2507.23279
>
> [3] Plan-and-Solve Prompting: Improving Zero-Shot Chain-of-Thought Reasoning by Large Language Models. ACL 2023.
>
> [4] Large Language Models as Optimizers. ICLR 2024.
>
> [5] https://huggingface.co/datasets/HuggingFaceH4/aime_2024
>
> [6]Measuring Faithfulness in Chain-of-Thought Reasoning arXiv:2307.13702
>
> [7]Think before you speak: Training Language Models With Pause Tokens arXiv:2310.02226
>
> [8]Learning to Insert [PAUSE] Tokens for Better Reasoning 2506.03616
>
> [9]Pause Tokens Strictly Increase the Expressivity of Constant-Depth Transformers arXiv:2505.21024
>
> [10] GVPO: Group Variance Policy Optimization for Large Language Model Post-Training arXiv:2504.19599
>
> [11] BNPO: Beta Normalization Policy Optimization arXiv:2506.02864
>
> [12] https://huggingface.co/deepseek-ai/DeepSeek-R1-Distill-Qwen-7B
>
> [13] https://huggingface.co/deepseek-ai/deepseek-math-7b-instruct

---

> > ### Comment · Reviewer_z6zV · 2025-11-26
> >
> > 1) The authors indeed agree that the later layers may compensate for the phenomenon they observe in layer 1 (Reply 2). This really makes it unclear whether the current motivation for ARM makes sense.
> >
> > 2) The sensitivity analysis in Appendix F highlights that gains come for extremely small and non-trivial choice of hyperparameters. In most cases, gains are <1%.
> >
> > 3) More importantly, the Table B results added above do not make sense. The authors mention that they reduced the max generation length to 512. But clearly, this has caused a massive drop in performance. Pass@1 with generation lenght of 2048 is 63.9 (Table 1). However, with gen length of 512, even Pass@4 is just 48.0. These results thus make it probably clear that meaningless tokens don't really matter if sufficiently long (and required) max generation length is give for CoT.
> >
> > 4) Temperature: There is no default temperature value for a LLM. You are supposed to do a thorough temperature search at each pass@k to find the optimal setting. For k=0, generally temperature = 0 works the best. For higher K's, it varies.
> >
> >
> > **To summarize**
> > Given major issues in experimental setups of this manuscript, I maintain my rating. I thank the authors for their rebuttal.

---

> > > ### Author Response · Authors · 2025-11-28
> > > **Reply (1/2)**
> > >
> > > >***1. The authors indeed agree that the later layers may compensate for the phenomenon they observe in layer 1 (Reply 2). This really makes it unclear whether the current motivation for ARM makes sense.***
> > >
> > >
> > > - We would like to clarify that for a 30+ layer LLM, it is actually expected and normal that later layers may compensate, amplify, or override transformations occurring in the first layer. What would be unexpected is the opposite, namely, that a modification in the first layer has no systematic effect on the representations processed by subsequent layers. Our results clearly show that applying the affine transformaton in the first layer leads to a **consistently different transformation trajectory across later layers compared to the no shift condition, indicating that the first layer change is propagated and utilized, rather than being fully compensated**.
> > >
> > > - Moreover, our multi-layer ARM experiments strongly support our motivation: when ARM is applied both to the first layer and many later layers, the model produces normal, coherent outputs; however, when ARM is applied only to many later layers while ignoring the first layer, the model outputs nonsensical text. This contrast demonstrates that the first layer plays a critical anchoring role, and that ARM must be applied there for the model to behave properly. Therefore, the observed compensation behavior does not undermine the motivation of ARM; instead, it highlights the necessity of incorporating ARM at the first layer, which is precisely what our method leverages.
> > >
> > >
> > > ||Answer/Pass@1|
> > > |-|-|
> > > |Every Layers|**garbled text**|
> > > |Baseline|63.8|
> > > |Every 3 Layers (w/ Layer1)|64.0|
> > > |Every 3 Layers(w/o Layer1)|**garbled text**|
> > > |Every 5 Layers(w/ Layer1)|64.8|
> > > |Every 5 Layers(w/o Layer1)|**garbled text**|
> > > |Every 8 Layers(w/ Layer1)|66.0|
> > > |Every 8 Layers(w/o Layer1)|**garbled text**|
> > > |ARM|67.0|
> > >
> > >
> > >
> > > >***2.The sensitivity analysis in Appendix F highlights that gains come for extremely small and non-trivial choice of hyperparameters. In most cases, gains are <1%.***
> > >
> > >
> > > - We believe there is a misunderstanding regarding Appendix F. The examples where the gains are <1% do not indicate a small and non-trivial hyperparameter choice, but rather reflect **settings outside the optimal range**. As clearly stated in the appendix, the purpose of this experiment is to show that the optimal region is concentrated around a stable interval, e.g., c in [0.25, 0.30] and p in [85,92.5] Within this range, our method consistently yields notable improvements, demonstrating that the hyperparameter selection is robust and does not rely on a single carefully picked value.
> > >
> > > - Therefore, the <1% gains only confirm that those settings are **not within the optimal range**, rather than supporting the claim that our method depends on a small and non-trivial choice of hyperparameters.
> > >
> > >
> > > >***3.More importantly, the Table B results added above do not make sense. The authors mention that they reduced the max generation length to 512. But clearly, this has caused a massive drop in performance. Pass@1 with generation lenght of 2048 is 63.9 (Table 1). However, with gen length of 512, even Pass@4 is just 48.0. These results thus make it probably clear that meaningless tokens don't really matter if sufficiently long (and required) max generation length is give for CoT.***
> > >
> > > - We believe there is a misunderstanding regarding the purpose of Table B. As clearly stated in the rebuttal, we reduced the maximum generation length to 512 due to computational constraints, **in order to isolate and verify the effectiveness of meaningless tokens under a fixed inference budget**. It is well established that allowing longer generation generally improves CoT performance, **but this improvement comes with a significant time–latency trade-off, as longer max lengths substantially increase decoding time**.
> > >
> > > - Our key point is that meaningless tokens can improve performance without increasing inference time, and both Table 1 (with max length = 2048) and Table B (with max length = 512) **consistently show performance gains under their respective generation budgets**. Therefore, the lower absolute scores in Table B simply reflect the shorter generation length, rather than indicating that meaningless tokens become ineffective. On the contrary, the consistent improvements across different max lengths demonstrate that meaningless tokens remain beneficial even when additional generation length is not available, which is crucial for practical, time-constrained settings.

---

> > > ### Author Response · Authors · 2025-11-28
> > > **Reply(2/2)**
> > >
> > > >***4.Temperature: There is no default temperature value for a LLM. You are supposed to do a thorough temperature search at each pass@k to find the optimal setting. For k=0, generally temperature = 0 works the best. For higher K's, it varies.***
> > >
> > > - We would like to clarify a misunderstanding regarding the notion of default temperature. When the temperature is not explicitly specified in model.generate, most LLMs do fall back to a default value defined either in their generation_config.json (e.g., Qwen2.5-7B-Instruct uses temperature = 0.7) or in the Transformers library defaults. This is the default temperature we refer to, and it is the value used when practitioners call the model without manually setting temperature.
> > >
> > > - Following the your suggestion, we have additionally reported results for pass@4, 8, and 16 with ARM, as well as experiments under different temperature settings. These results consistently show improvements, demonstrating that ARM provides benefits across pass@k and temperature variations. Importantly, conducting a thorough temperature search for each pass@k is not aligned with our main claim. **Our goal is not to identify the optimal temperature for each evaluation setting, but to show that ARM yields robust performance gains without relying on careful hyperparameter tuning.** A full temperature sweep would only optimize absolute performance, but would not provide evidence relevant to our central claim that ARM consistently improves performance under diverse and practical inference configurations. Therefore, we believe that such an exhaustive search is unnecessary for supporting our conclusions.

---

### Official Review · Reviewer_rUSd · 2025-11-01

**Soundness:** 2
**Presentation:** 3
**Contribution:** 2
**Rating:** 4
**Confidence:** 3

**Summary:**

The authors observe a puzzling phenomenon—the "meaningless-token effect"—and find that prepending meaningless tokens to the prompt induces a shift in the activation distribution of the first-layer MLP. Building on this observation, they propose a training-free, inference-time activation redistribution strategy that effectively enhances performance on reasoning tasks.

**Strengths:**

1. The paper is well-written, clear, and easy to follow.
2. It provides an in-depth analysis of an unusual phenomenon, offering a thoughtful and insightful perspective.
3. The observation that inserting meaningless tokens reduces the proportion of near-zero activations is compelling, and the paper supports this with solid theoretical analysis.

**Weaknesses:**

1. Table 1 raises significant concerns: the authors' reproduced results for these models are inconsistent with—and notably lower than—the results reported in the original papers. This discrepancy casts doubt on the completeness and accuracy of the evaluation methodology used in this work.
2. The evidence for performance improvement on reasoning tasks is insufficient, as the reported results for all baselines are lower than expected. Moreover, the paper lacks experiments with large reasoning models, leaving it unclear how the proposed method performs on LRMs.
3. Figures 2 and 3 both analyze the impact of inserting meaningless tokens, which is only tangentially related to the paper’s main conclusion. The authors’ explanation of this correlation relies primarily on Figure 7. However, Figure 7 merely states that activations for digits, operators, and logical connectives are more likely to be near zero, while other tokens exhibit the lowest proportion of near-zero activations. The authors seem to imply that their method may increase the activation magnitudes of digits, operators, and logical connectives. Yet, they do not provide the post-intervention activation distributions for these four token categories (digits, operators, logical connectives, and others), leaving the claimed mechanism unverified and the explanation incomplete.

**Questions:**

Please see the weaknesses.

---

> ### Author Response · Authors · 2025-11-20
> **Reply(1/3)**
>
> >***1. Table 1 raises significant concerns: the authors' reproduced results for these models are inconsistent with—and notably lower than—the results reported in the original papers. This discrepancy casts doubt on the completeness and accuracy of the evaluation methodology used in this work.***
>
> Thanks for your question.
>
>
> - The differences in evaluation metrics are primarily due to the **reasoning token length**. In our experiments, to ensure a fair comparison and given our computational constraints，we set **`max_length = 2048`** for all evaluations in Table 1. This setting differs from the maximum lengths used in the official technical reports of the respective models. Since a **larger max_length typically yields stronger performance**, some of our scores appear **lower than the reported values in the original papers**. We emphasize, however, that **all models were evaluated under identical conditions**, ensuring a **fair and consistent comparison** across the board. As shown in Table A, using Math-500 as an example, the discrepancies arise solely because our evaluation length is fixed at 2048, whereas the expected results in prior work were obtained using different (often larger) maximum lengths.
>
> - To further illustrate this effect, we use Qwen2.5-Math-1.5B and Qwen2.5-Math-7B as examples. In their original configuration, the maximum generation length is **512**. When we increase the **max_length to 2048**, the evaluation scores become **higher than the official results**, further demonstrating that response-length settings substantially influence measured model performance. The largest discrepancy appears in the DS-R1-Qwen-7B results. According to the official repository, this model uses a **maximum output length of 32,786**. Therefore, restricting the maximum output length to 2048 naturally leads to lower scores.
>
>
> **Table A. Expected results vs. our results (max_length = 2048).**
>
> |Math500|Original papers results|Max_Length|Ours results|Max Length(Ours)|
> | --- | --- | --- | --- | --- |
> |Qwen-2.5-Math-1.5b|49.8| 512|63.8|2048|
> |Qwen-2.5-Math-7b|55.4|512|72.4|2048|
> |DS-R1-Qwen-7B|92.8|32,768|52.7|2048|
> |DS-Math-7B-instruct|46.8|32,768|39.5|2048|
> |Llama-3.1-8B-Instruct|51.9|5120|41.8|2048|
> |Qwen-2.5-32B-Instruct|83.1|8192|81.3|2048|
>
>
> - We added an experiment to analyze the performance of DS-R1-Qwen-7B on Math-500 as the maximum output length increases in Table.B. The results show that the difference between our numbers and those in the technical report is primarily due to the max-length setting.
>
> **Table B. The relationship between performance and max length**
>
> |Max length|2048|4096|8192|
> | --- | --- | --- | --- |
> |DS-R1-Qwen-7B|52.7|72.8|83.4|

---

> ### Author Response · Authors · 2025-11-20
> **Reply(2/3)**
>
> > ***2. The evidence for performance improvement on reasoning tasks is insufficient, as the reported results for all baselines are lower than expected.***
>
>
> - Similarly to the Question1, for the results presented in Table2, the **max length of our work is different from the expected answers**. The experimental details are in Table C. We choose Math500 and Humaneval as examples. For Math500, we use max_length=2048, for Humaneval, we use the max_length=512, which is the default setting in its repo.
>
> **Table C. Expected results vs. our results with different max length on Math500 and Humaneval.**
>
> |Math500|Original papers results|Max Length|Ours results|Max Length(Ours)||HumanEval|Original papers results|Max Length|Ours results|Max Length(Ours)|
> | --- | --- | --- | --- | --- |--- | --- | --- | ---|---|---|
> |Qwen-2.5-Math-1.5B|49.8| 512|63.8|2048||Qwen-2.5-Math-1.5B|N/A|N/A|6.1|512|
> |Qwen-2.5-Math-7b|55.4|512|72.4|2048||Qwen-2.5-Math-7b|N/A|N/A|15.2|512|
> |Qwen-2.5-7b-Instruct|75.5|8192|73.2|2048||Qwen-2.5-7b-Instruct|52.9|8192|43.9|512|
> |Qwen-2.5-32B-Instruct|83.1|8192|82.6|2048||Qwen-2.5-32B-Instruct|60.97|8192|50.0|512|
> |Llama-3.1-8B-Instruct|51.9|5120|41.8|2048||Llama-3.1-8B-Instruct|72.6|1024|45.7|512|
> |gemma-3-4b-it|75.6|8192|73.6|2048||gemma-3-4b-it|71.3|8192|17.1|512|
> |gemma-3-27b-it|89.0|8192|85.4|2048||gemma-3-27b-it|87.8|8192|9.1|512|
>
>
> - **As the maximum generation length decreases, model performance naturally degrades**. The max length of our experiment is set lower than original papers because of the limitation in computational resources. However, in our experiments, all test sets are evaluated under the **same experimental configuration**, ensuring the completeness and accuracy of the evaluation methodology. We fully understand your concerns regarding the reported metrics. Nevertheless, due to limitations in computational resources, many prior works report results that may be higher or lower than the official numbers while still maintaining fair experimental conditions [1–9]. The examples we cite were selected arbitrarily, and we do not question the validity of their reported results. Our point is that **a fair and consistent evaluation setup is sufficient to verify the effectiveness of the proposed method**. That said, establishing a standardized evaluation protocol may indeed be an important direction for future research in this area. We appreciate your suggestions in this aspect.
>
>
>
>
> >***3. The paper lacks experiments with large reasoning models, leaving it unclear how the proposed method performs on LRMs.***
>
> Thank you for this valuable comment. We agree that evaluating on strong reasoning models is important. We would like to clarify that **our experiments already include representative Large Reasoning Models** (LRMs), following the definitions and experiments used in recent work.
>
> - **Models commonly considered LRMs in prior studies.** Prior work such as [4], [7], and [12] evaluates reasoning performance primarily using instruction-tuned or math-specialized models. These works identify models like Qwen2.5-Math, Qwen2.5-Instruct, and Llama-3.1-Instruct as standard representatives for LRM-based reasoning evaluation. Our experimental setup follows the same practice.
>
> - **LRMs defined in recent papers.** According to recent surveys [10, 11], the category of Large Reasoning Models includes the Qwen2.5-Math series, DeepSeek-Math series, and R1-Distill series. In our paper (Table 1 and Table 2), we evaluate precisely these models: Qwen2.5-Math-1.5B, Qwen2.5-Math-7B, DeepSeek-Math-7B-Instruct, DeepSeek-R1-Distill-Qwen-7B. These models are widely recognized as LRMs designed or adapted specifically for mathematical and logical reasoning.
>
> - **Our method demonstrates consistent gains across LRMs.** We report improvements across all of the above LRM families. For instance, both Qwen2.5-Math-1.5B and Qwen2.5-Math-7B show clear and consistent performance gains using our method, supporting its applicability to LRMs of different sizes.
>
> In summary, while we appreciate the reviewer’s concern, our experiments already include multiple representative Large Reasoning Models, and the results confirm that our method generalizes well across these widely used LRMs.

---

> ### Author Response · Authors · 2025-11-20
> **Reply(3/3)**
>
> >***4. Figures 2 and 3 both analyze the impact of inserting meaningless tokens, which is only tangentially related to the paper’s main conclusion. The authors’ explanation of this correlation relies primarily on Figure 7. However, Figure 7 merely states that activations for digits, operators, and logical connectives are more likely to be near zero, while other tokens exhibit the lowest proportion of near-zero activations. The authors seem to imply that their method may increase the activation magnitudes of digits, operators, and logical connectives. Yet, they do not provide the post-intervention activation distributions for these four token categories (digits, operators, logical connectives, and others), leaving the claimed mechanism unverified and the explanation incomplete.***
>
> We appreciate you pointing this out. According to your suggestion, we follow the experiments in Figure.7 and computed the **average activation sparsity** for each token category before and after the affine transformation induced by the insertion of meaningless tokens on Qwen2.5-Math-7B and Llama3.1-8B-Instruct. From Table.D, we observe that the activation sparsity of digit, operator, and conjunction tokens decreases, while the sparsity of other tokens increases. This pattern supports our explanation: the affine transformation induced by the meaningless tokens **enhances the activation strength of tokens that are crucial for reasoning** (digits, operators, conjunctions), while suppressing less relevant tokens.
>
>
> **Table D. Average activation sparsity w/ Meaningless tokens vs. w/o Meaningless tokens**
>
> ||digit|operator|conjunction|other|
> |---|---|---|---|---|
> |Qwen2.5-Math-7B|6.69 -> 5.52|5.34 -> 5.31|5.27 -> 5.20|4.85 ->5.14|
> |Llama3.1-8B-Intruct|10.19->9.78|13.94->9.88|11.15->9.90|9.43->9.91|
>
>
>
>
> [1]Angles Don’t Lie: Unlocking Training-Efficient RL Through the Model’s Own Signals arXiv:2506.02281
>
> [2]PICSAR: Probabilistic Confidence Selection and Ranking For Reasoning Chains. arXiv:2508.21787
>
> [3]CoT-Self-Instruct: Building high-quality synthetic prompts for reasoning and non-reasoning tasks arXiv:2507.23751
>
> [4]Enhancing Test-Time Scaling of Large Language Models with Hierarchical Retrieval-Augmented MCTS arXiv:2507.05557
>
> [5]MOTIF: Modular Thinking via Reinforcement Fine-tuning in LLMs arXiv:2507.02851
>
> [6]DuaShepherd: Integrating Stepwise Correctness and Potential Rewards for Mathematical Reasoning arXiv:2506.17533
>
> [7]ReasonFlux: Hierarchical LLM Reasoning via Scaling Thought Templates arXiv:2502.06772
>
> [8]Confidence Is All You Need: Few-Shot RL Fine-Tuning of Language Models arXiv:2506.06395
>
> [9]Controlling Large Language Model with Latent Actions arXiv:2503.21383
>
> [10] Towards Large Reasoning Models: A Survey of Reinforced Reasoning with Large Language Models arXiv:2501.09686
>
> [11] A Survey of Reinforcement Learning for Large Reasoning Modelsh arXiv:2509.08827
>
> [12] Unveiling Super Experts in Mixture-of-Experts Large Language Models arXiv:2507.23279

---

> ### Author Response · Authors · 2025-11-28
> **Follow up on Our Previous Response**
>
> Thank you again for your thoughtful comments. We would like to note that we have thoroughly addressed all of your concerns:
>
> - We listed several discrepancies between the official technical reports and our actual experimental settings. We showed that **our results can be either higher or lower than the official reports**, and we also provided examples from recent papers to demonstrate that results may naturally differ from those reported in the official documentation. Moreover, we ensured that **all of our experimental settings were fully fair and controlled, which further highlights the effectiveness of our method**.
>
> - We provided **examples from recent reasoning focused works and clarified the definition of LRM as described in the survey**. This shows that our experiments indeed follow the standard LRM evaluation setup.
>
> - In response to your third question, we have **added an analysis of the proportion changes across different token types** after introducing meaningless tokens that induce an affine transformation. This further demonstrates that the affine transformation does increase the activation proportion of key tokens like digit, cooperation and conjunction.
>
> If you have any additional questions or need further clarification, we would be more than happy to provide it.

---

### Meta-Review · Area_Chair_vuPP · 2026-01-08

**Summary:**

This paper investigated an interesting and counterintuitive phenomenon: inserting meaningless tokens between the user query and system prompt can benefit LLMs reasoning. This work presented the Activation Redistribution Module (ARM), an inference-time method that directly intervene activations without modifying inputs. Empirical results across diverse models and benchmarks demonstrated that ARM can consistently improve LLMs on reasoning tasks.

While this work is quite interesting and effective, there are several concerns proposed by the reviewer stand quite important:
1. The comparison is not convincing enough, as most of the benchmark baselines are notably lower than the results in their original papers. Although the authors argued in the rebuttals that this is because they want to keep all the evaluation in the same page and used a universal length (2048), I do not quite buy this explanation. Since for many reasoning tasks, such length does not guarantee the good enough performance, and improving the performance from a significantly lower level is much easier. What we want to see in this community is the contribution that helps the best performance getting better (if a method claims it brings "gains").
2. If the first point is difficult to achieve, there are still enough value if a research reveals some deep insights. This paper proposed inserting meaningless tokens between query and system prompt to improve the reasoning performance, which is quite interesting observation. However, the authors did not provide enough theoretical analysis to bring us more insights into this phenomenon (the analysis regarding the variance shift is relative superficial), and the only meaningless token studied are '/' and '?' and the behavior studied is only repetition, though the authors provide some more research in table E, the scope of the recipe is still narrow.
3. Combining the first and second point, and the fact that there are relatively sensitive hyper-parameters (Appendix F.7 does not lead a "robust and universal HP selection" conclusion), it leads to a frustrating truth - that there is not a solid and robustly effective recipe for users   to actually launch this method in their model if they have model improvement requirement.

Overall, I actually like the observation and idea proposed in this paper, and I do think this is quite interesting, but it's current form just does not fulfill a venue like ICLR. I hope the authors could take the suggestions and opinions from the reviewers and further optimize the current research, which I believe will be valuable enough in the future.

**Reviewer Concerns:**

Many of the meaningful concerns were actually addressed by the rebuttal, however, please see above summary to find that there are still outstanding concerns.

**Reviewer Scores:**

The original scores are 2(z6ZV), 6(n5fR), 4(3ry9) and 4(rUSd). I would expect a change to 2,6,6,4.

---

### Decision · Program_Chairs · 2026-01-26

Reject